# Mixture-of-Experts Variational Autoencoder for Clustering and Generating from Similarity-Based Representations

## Abstract

Clustering high-dimensional data, such as images or biological measurements, is a long-standing problem and has been studied extensively. Recently, Deep Clustering gained popularity due to the non-linearity of neural networks, which allows for flexibility in fitting the specific peculiarities of complex data. Here we introduce the Mixture-of-Experts Similarity Variational Autoencoder (MoE-Sim-VAE), a novel generative clustering model. The model can learn multi-modal distributions of high-dimensional data and use these to generate realistic data with high efficacy and efficiency. MoE-Sim-VAE is based on a Variational Autoencoder (VAE), where the decoder consists of a Mixture-of-Experts (MoE) architecture. This specific architecture allows for various modes of the data to be automatically learned by means of the experts. Additionally, we encourage the lower dimensional latent representation of our model to follow a Gaussian mixture distribution and to accurately represent the similarities between the data points. We assess the performance of our model on synthetic data, the MNIST benchmark data set, and a challenging real-world task of defining cell subpopulations from mass cytometry (CyTOF) measurements on hundreds of different datasets. MoE-Sim-VAE exhibits superior clustering performance on all these tasks in comparison to the baselines and we show that the MoE architecture in the decoder reduces the computational cost of sampling specific data modes with high fidelity.

## 1 Introduction

Clustering has been studied extensively (Aljalbout et al., 2018; Min et al., 2018) in machine learning. Recently, many Deep Clustering approaches were proposed, which modified (Variational) Autoencoder ((V)AE) architectures (Min et al., 2018; Zhang et al., 2017) or with varying regularization of the latent representation (Dizaji et al., 2017; Jiang et al., 2017; Yang et al., 2017; Fortuin et al., 2019).

Reconstruction error usually drives the definition of the latent representation learned from an AE or VAE. The representation for AE models is unconstrained and typically places data objects close to each other according to an implicit similarity measure that also yields favorable reconstruction error. In contrast, VAE models regularize the latent representation such that the represented inputs follow a certain variational distribution. This construction enables sampling from the latent representation and data generation via the decoder of a VAE. Typically, the variational distribution is assumed standard Gaussian, but for example, Jiang et al. (2017) introduced a mixture of Gaussian variational distribution for clustering purposes.

A key component of clustering approaches is the choice of similarity metric for the considered data objects which we try to group (Irani et al., 2016). Such similarity metrics are either defined *a priori* or learned from the data to specifically solve classification tasks via a Siamese network architecture (Chopra et al., 2005). Dimensionality reduction approaches, such as UMAP (McInnes et al., 2018) or t-SNE (van der Maaten & Hinton, 2008), allow to specify a similarity metric for projection and thereby define the data separation in the inferred latent representation.

In this work, we introduce the Mixture-of-Experts Similarity Variational Autoencoder (MoE-Sim-VAE), a new deep architecture that performs similarity-based representation learning, clustering of the data and generation of data from each specific data mode. Due to a combined loss function,

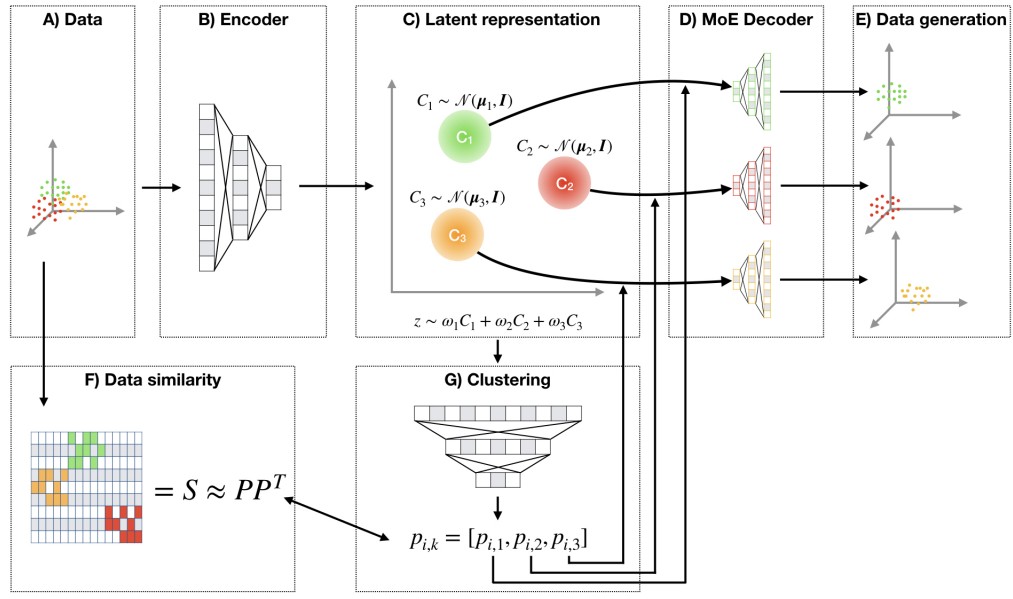

Figure 1: Overview of the proposed model MoE-Sim-VAE. Data (in panel A) gets encoded via a encoder network (B) into a latent representation (C) which is trained to be a mixture of standard Gaussians. Via a clustering network (G), which is trained to reconstruct a user-defined similarity matrix (F), the encoded samples get assigned to the data mode-specific decoder subnetworks (which we call experts) in the MoE Decoder (D). The experts reconstruct the original input data and can be used for data generation when sampling from the variational distribution (E).

it can be jointly optimized. We assess the scope of the model on synthetic data and we present superior clustering performance on MNIST. Moreover, in an ablation study, we show the efficiency and precision of MoE-Sim-VAE for data generation purposes in comparison to the most related state-of-the-art method (Jiang et al., 2017). Finally, we show an application of MoE-Sim-VAE on a real-world clustering problem in biology on multiple datasets.

Our main contributions are to

- Develop a novel autoencoder architecture for
    - similarity-based representation learning
    - unsupervised clustering
    - accurate and efficient data generation
- Embed the Mixture-of-Expert architecture into a Variational Autoencoder setup to train a separate generator for each data mode
- Show superior clustering performance of the model on benchmark dataset and real-world biological data

## 2 MIXTURE-OF-EXPERTS SIMILARITY VARIATIONAL AUTOENCODER

Here we introduce the Mixture-of-Experts Similarity Variational Autoencoder (MoE-Sim-VAE, Figure 1). The model is based on the Variational Autoencoder (Kingma & Welling, 2014). While the encoder network is shared across all data points, the decoder of the MoE-Sim-VAE consists of a number of $K$ different subnetworks, forming a Mixture-of-Experts architecture (Shazeer et al., 2017). Each subnetworks constitutes a generator for a specific data mode and is learned from the data.

The variational distribution over the latent representation is defined to be a mixture of multivariate Gaussians, first introduced by Jiang et al. (2017). In our model, we aim to learn the mixture components in the latent representation to be standard Gaussians

$$z \sim \sum_{k=0}^{K} \omega_k \mathcal{N}(\boldsymbol{\mu_k}, \boldsymbol{I}) \tag{1}$$

where $\omega_k$ are mixture coefficients, $\boldsymbol{\mu}_k$ are the means for each mixture component, $\boldsymbol{I}$ is the identity matrix and $K$ is the number of mixture components. The dimension of the latent representation $z$ needs to be defined to suit the demands of Gaussian mixtures which have limitations in higher dimensions (Bishop, 1995). Similar to optimizing an Evidence Lower Bound (ELBO), we penalize the latent representation via the reconstruction loss of the data $\mathcal{L}_{reconst}$ and by using the Kullback-Leibler (KL) divergence for multivariate Gaussians (Jiang et al., 2017) on the latent representation

$$\mathcal{L}_{KL} = D_{KL}(\mathcal{N}_0, \mathcal{N}_1) = \frac{1}{2}\{tr(\boldsymbol{\Sigma}_1^{-1}\boldsymbol{\Sigma}_0) + (\boldsymbol{\mu}_1 - \boldsymbol{\mu}_0)^T\boldsymbol{\Sigma}_1^{-1}(\boldsymbol{\mu}_1 - \boldsymbol{\mu}_0) - k + ln\frac{|\boldsymbol{\Sigma}_1|}{|\boldsymbol{\Sigma}_0|}\} \quad (2)$$

where $k$ is a constant, $\mathcal{N}_0 \sim \mathcal{N}(\boldsymbol{\mu}_0, \boldsymbol{\Sigma}_0 = \boldsymbol{I})$, and $\boldsymbol{I}$ is the identity matrix. Further, $\mathcal{N}_1 \sim \mathcal{N}(\boldsymbol{\mu}_1, \boldsymbol{\Sigma}_1 = diag(\sigma_j))$, where $\sigma_j$ for $j = 1, \ldots, D$, for a number of dimensions $D$, is estimated from the samples of the latent representation. Finally, we assume $\boldsymbol{\mu}_0 = \boldsymbol{\mu}_1$ resulting in the following simplified objective

$$\mathcal{L}_{KL} = D_{KL}(\mathcal{N}_0, \mathcal{N}_1) = \frac{1}{2}\{tr(\boldsymbol{\Sigma}_1^{-1}\boldsymbol{\Sigma}_0) - k + ln\frac{|\boldsymbol{\Sigma}_1|}{|\boldsymbol{\Sigma}_0|}\} \, , \quad (3)$$

to penalize exclusively the covariance of each cluster. It remains to define the reconstruction loss $\mathcal{L}_{reconst}$, where we choose a Binary Cross-Entropy

$$\mathcal{L}_{reconst} = \sum_i^N \sum_d^D x_{i,d} \log(x_{i,d}^{reconst}) \quad (4)$$

between the original data $x$ (scaled between 0 and 1) and the reconstructed data $x^{reconst}$, where $i$ iterates the batch size $N$ and $d$ the dimensions of the data $D$. Finally the loss for the VAE part is defined by

$$\mathcal{L}_{VAE} = \mathcal{L}_{reconst} + \pi_1 \mathcal{L}_{KL} \quad (5)$$

with a weighting coefficient $\pi_1$ which can be optimized as a hyperparameter.

### SIMILARITY CLUSTERING AND GATING OF LATENT REPRESENTATION

Training of a data mode-specific generator expert requires samples from the same data mode. This necessitates to solve a clustering problem, that is, mapping the data via the latent representation into $K$ clusters, each corresponding to one of the $K$ generator experts. We solve this clustering problem via a clustering network, also referred to as gating network for MoE models. It takes as input the latent representation $\boldsymbol{z}_i$ of sample $i$ and outputs probabilities $p_{ik} \in [0, 1]$ for clustering sample $i$ into cluster $k$. According to this cluster assignment, sample $i$ is then gated to expert $k = \mathrm{argmax}_k p_{ik}$ for each sample $i$. We further define the cluster centers $\boldsymbol{\mu}_k$ for $k \in \{1, \ldots, K\}$ similar as in the Expectation Maximization (EM) algorithm for Gaussian Mixture models (Bishop, 2006) as

$$\boldsymbol{\mu}_k = \frac{1}{N_k} \sum_{i=1}^N p_{ik} \boldsymbol{z}_i \, , \quad (6)$$

where $N_k$ is the absolute number of data points assigned to cluster $k$ based on highest probability $p_{ik}$ for each sample $i = 1, \ldots, N$. The Gaussian mixture distributed latent representation (via KL loss in Equation 3) is motivation for the empirical computation of the cluster means and further, similar as in the EM algorithm, it allows iterative optimization of the means of the Gaussians. We train the clustering network to reconstruct a data-driven similarity matrix $\boldsymbol{S}$, using the Binary Cross-Entropy

$$\mathcal{L}_{Similarity} = \sum_i^N \sum_j^N \boldsymbol{S}_{i,j} \log((\boldsymbol{P}\boldsymbol{P}^T)_{i,j}) \quad (7)$$

to minimize the error in $\boldsymbol{P}\boldsymbol{P}^T \approx \boldsymbol{S}$, with $\boldsymbol{P} := \{p_{ik}\}_{i \in \{1, \ldots, N\}, k \in \{1, \ldots, K\}}$ where $N$ is the number of samples (e.g., batch size). Intuitively, $\boldsymbol{P}\boldsymbol{P}^T$ approximates the similarity matrix $\boldsymbol{S}$ since values in $\boldsymbol{P}\boldsymbol{P}^T$ are only close to 1 when similar data objects are assigned to the same cluster, similar to the entries in the adjacency similarity matrix $\boldsymbol{S}$. This similarity matrix is derived in an unsupervised way in our experiments (e.g. UMAP projection of the data and k-nearest-neighbors or distance

thresholding to define the adjacency matrix for the batch), but can also be used to include weakly-supervised information (e.g., knowledge about diseased vs. non-diseased patients). If labels are available, the model could even be used to derive a latent representation with supervision. The similarity feature in MoE-Sim-VAE thus allows to include prior knowledge about the best similarity measure on the data.

Moreover, we apply the DEPICT loss from Dizaji et al. (2017), to improve the robustness of the clustering. For the DEPICT loss, we additionally propagate a noisy probability $\hat{p}_{ik}$ through the clustering network using dropout after each layer. The goal is to predict the same cluster for both, the noisy $\hat{p}_{ik}$ and the clean probability $p_{ik}$ (without applying dropout). Dizaji et al. (2017) derived as objective function a standard cross-entropy loss

$$\mathcal{L}_{DEPICT} = -\frac{1}{N} \sum_{i=0}^{N} \sum_{k=0}^{K} q_{ik} \log(\hat{p}_{ik}) \tag{8}$$

whereby $q_{ik}$ is computed via the auxiliary function

$$q_{ik} = \frac{p_{ik}/(\sum_{i'} p_{i'k})^{\frac{1}{2}}}{\sum_{k'} p_{ik'}/(\sum_{i'} p_{i'k'})^{\frac{1}{2}}} \tag{9}$$

where we refer to Dizaji et al. (2017) for exact derivation. The DEPICT loss encourages the model to learn invariant features from the latent representation for clustering with respect to noise (Dizaji et al., 2017). Looking at it from a different perspective, the loss helps to define a latent representation which has those invariant features to be able to reconstruct the similarity and therefore the clustering correctly. The complete clustering loss function $\mathcal{L}_{Clustering}$ is then defined by

$$\mathcal{L}_{Clustering} = \mathcal{L}_{Similarity} + \pi_2 \mathcal{L}_{DEPICT} \tag{10}$$

with a mixture coefficient $\pi_2$ which can be optimized as a hyperparameter.

MoE-Sim-VAE loss function

Finally, the MoE-Sim-VAE model loss is defined by

$$\mathcal{L}_{MoE-Sim-VAE} = \underbrace{\mathcal{L}_{VAE}}_{\mathcal{L}_{reconst}+\pi_1 \mathcal{L}_{KL}} + \underbrace{\mathcal{L}_{Clustering}}_{\mathcal{L}_{Similarity}+\pi_2 \mathcal{L}_{DEPICT}} \tag{11}$$

which consists of the two main loss functions $\mathcal{L}_{VAE}$, acting as a regularization for the latent representation, and $\mathcal{L}_{Clustering}$, which helps to learn the mixture components based on an *a priori* defined data similarity. The model objective function $\mathcal{L}_{MoE-Sim-VAE}$ can then be optimized end-to-end to train all parts of the model.

## 3 RELATED WORK

(V)AEs have been extensively used for clustering (Xie et al., 2016; Dizaji et al., 2017; Li et al., 2017; Yang et al., 2017; Saito & Tan, 2017; Chen et al., 2017; Aljalbout et al., 2018; Fortuin et al., 2019). The most related approaches to MoE-Sim-VAE are Jiang et al. (2017) and Zhang et al. (2017).

Jiang et al. (2017) introduced the VaDE model, comprising a mixture of Gaussians as underlying distribution in the latent representation of a Variational Autoencoder. Optimizing the Evidence Lower Bound (ELBO) of the log-likelihood of the data can be rewritten to optimize the reconstruction loss of the data and KL divergence between the variational posterior and the mixture of Gaussians prior. Jiang et al. (2017) motivate the use of to two separate networks for reconstruction and the generation process of the model. Further, to effectively generate images from a specific data mode and to increase image quality, the sampled points have to surpass a certain posterior threshold and are otherwise rejected. This leads to an increased computational effort. The MoE Decoder of our model, which is used for both reconstruction and generation, does not need such a threshold, as we discuss in more detail in Section 4.2.1.

Zhang et al. (2017) have introduced a mixture of autoencoders (MIXAE) model. The latent representation of the MIXAE is defined as the concatenation of the latent representation vectors of each single autoencoder in the model. Based on this concatenated latent representation, a Mixture

Table 1: Performance comparison of our method MoE-Sim-VAE with several published methods. The Table is mainly extracted from Aljalbout et al. (2018) and complemented with results of interest. (" - ": metric not reported)

| METHOD | NMI | ACC |
|---|---|---|
| JULE, Yang et al. (2016b) | 0.915 | - 10 |
| CCNN, Hsu & Lin (2017) | 0.876 | - |
| DEC, Xie et al. (2016) | 0.8 | 0.843 |
| DBC, Li et al. (2017) | 0.917 | 0.964 |
| DEPICT, Dizaji et al. (2017) | 0.916 | 0.965 |
| DCN, Yang et al. (2017) | 0.81 | 0.83 |
| Neural Clustering, Saito & Tan (2017) | - | 0.966 |
| UMMC, Chen et al. (2017) | 0.864 | - |
| VaDE, Jiang et al. (2017) | - | 0.945 |
| TAGnet, Wang et al. (2016) | 0.651 | 0.692 |
| IMSAT, Hu et al. (2017) | - | **0.984** |
| Aljalbout et al. (2018) | 0.923 | 0.961 |
| MIXAE, Zhang et al. (2017) | - | 0.945 |
| Spectral clustering, Shaham et al. (2018) | 0.754 | 0.717 |
| SpectralNet (input space, Euclidean dist.), Shaham et al. (2018) | 0.687±0.004 | 0.622±0.008 |
| SpectralNet (input space, Siamese dist.), Shaham et al. (2018) | 0.884±0.02 | 0.826±0.03 |
| SpectralNet (code space, Euclidean dist.), Shaham et al. (2018) | 0.814±0.008 | 0.800±0.003 |
| SpectralNet (code space, Siamese dist.), Shaham et al. (2018) | 0.924±0.001 | 0.971±0.001 |
| *MoE-Sim-VAE* (proposed) | **0.935** | 0.975 |

Assignment Network predicts probabilities which are used in the Mixture Aggregation to form the output of the generator network. Each AE model learns the manifold of a specific cluster, similarly to our MoE Decoder. However, MIXAE does not optimize a variational distribution, such that generation of data from a distribution over the latent representation is not possible, in contrast to the MoE-Sim-VAE (Figure 2).

## 4 EXPERIMENTS

We evaluate the MoE-Sim-VAE using synthetic data and the MNIST data set of handwritten digits (LeCun et al., 1998) for clustering and data generation. Furthermore, we performed an ablation study to demonstrate the importance of the MoE Decoder. Finally, we present experiments on a real-world application of defining cellular subpopulations from mass cytometry measurements (Bandura et al., 2009) of multiple publicly available datasets (Weber & Robinson, 2016; Bodenmiller et al., 2012). Model implementation details are reported in the appendix in section A.1

We found that our model achieves superior clustering performance compared to other models on synthetic, MNIST and real-world datasets. Moreover, we show that MoE-Sim-VAE can more effectively and efficiently generate data from specific modes in comparison to other methods.

### 4.1 EVALUATION OF MOE-SIM-VAE ON SYNTHETIC DATA

We evaluated our model using data sampled from a 100-dimensional multivariate Gaussian with equal mixture weights for each component. We tested two aspects of our model: Firstly, we evaluated up to how many clusters our model can fit well. Therefore, we sampled data from distributions with up to a hundred mixture components. For this experiment, we assume knowledge of the true number of clusters in the data for both methods, MoE-Sim-VAE and GMMs. Secondly, we tested if our model is able to identify the true number of clusters in the data. The similarity matrix $S$ was defined as an adjacency matrix over the data items. Adjacency indicators were based on projecting the data via dimensionality reduction with UMAP (McInnes et al., 2018) and selecting neighbors according to a distance threshold. Details on model parameters can be found in Section A.1.1.

MoE-Sim-VAE performs better or comparable to the baseline for the number of clusters of up to 40 (Figure A1a). The model predicts with a close to perfect F-measure until reaching a true number

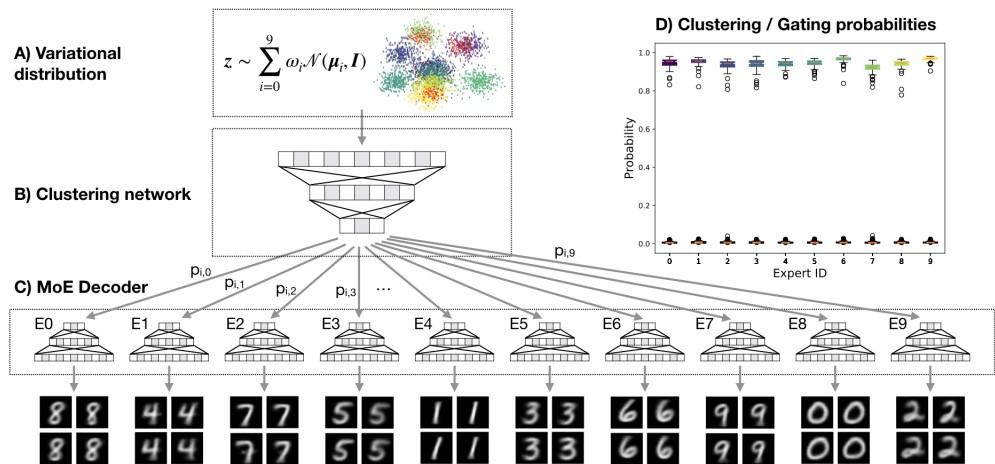

Figure 2: Generation of MNIST digit images. Data points from the latent representation were sampled from the variational distribution (A) which is learned to be a mixture of standard Gaussians and then clustered and gated (B) to the data-mode-specific experts of the MoE Decoder (C). (D) All samples from the variational distribution were correctly classified and therefore also correctly gated.

of clusters of 30. Within the range of true number of clusters from 30 to 40, the model performs comparable to GMMs. Further, MoE-Sim-VAE learns the true number of clusters on its own (Figure A1b). For up to 23 components in the data, MoE-Sim-VAE learns the true number of clusters even when defining a model with $K = 40$ experts in the MoE Decoder. This suggests that the model is robust to misspecification regarding the number of experts.

### 4.2 Unsupervised clustering, embedding and data generation of MNIST

We trained a MoE-Sim-VAE model on images from MNIST. We compared our model against multiple models which were recently reviewed in Aljalbout et al. (2018), and specifically against VaDE (Jiang et al., 2017) which shares similar properties with MoE-Sim-VAE (see Sec 3).

We compare the models with the Normalized Mutual Information (NMI) criterion but also classification accuracy (ACC) (Table 1). The MoE-Sim-VAE outperforms the other methods w.r.t. clustering performance when comparing NMI and achieves the second-best result when comparing ACC. Note that we used the number of experts $k = 10$ in our model to fit the existing number of digits in MNIST. Regarding the similarity measure, we decided to use as similarity a UMAP projection (McInnes et al., 2018) of MNIST and then apply k-nearest-neighbors of each sample in a batch. More details on the model are reported in Section A.1.2. In an ablation study we show the importance of the similarity matrix to create a clear separation of the different digits in the latent representation (Figure A4)

In addition to the clustering network, we can make use of the latent representation for image generation purposes. The latent representation is trained as a mixture of standard Gaussians. The means of these Gaussians are the centers of the clusters trained via the clustering network. Therefore, the variational distribution can be sampled from and gated to the cluster-specific expert in the MoE-decoder. The expert then generates new data points for the specific data mode. Results and the schematic are displayed in Figure 2 and in more detail and with greater sample size in the Appendix in Figure A2.

#### 4.2.1 Why does a MoE Decoder actually matter?

In an ablation study, we compare the two models MoE-Sim-VAE and VaDE (Jiang et al., 2017) on generating MNIST images with the request for a specific digit. The goal is to show that a MoE decoder, as proposed in our model, is beneficial. We focus our comparison to VaDE since this model, as the MoE-Sim-VAE, resorts to a mixture of Gaussian latent representation but differs in generating images by means of a single decoder network instead of a Mixture-of-Expert decoder network. The rationale for our design choice is to ensure that smaller sub-networks learn to reproduce and generate specific modes of the data, in this case of specific MNIST digits.

To show that both models' latent representations are separating the different clusters well, we computed the Maximum Mean Discrepancy (MMD), defined in Section A.1.2. The MMD can be inter-

preted as a distance between distributions computed based on samples drawn from these distributions. The heatmaps of the MMDs for VaDE and MoE-Sim-VAE as well as an UMAP projection of the latent representation colored with the mixture component confirm visually the separation of the clusters in the latent representations of both models (Fig. A3). As a result, we can conclude that both latent representations can separate the clusters of respective digits well, such that the decoder gets well-defined samples to generate the requested digit. Therefore, the main difference of generating specific digits arises in the decoder/generator networks.

We evaluated the importance of the MoE-Decoder to (1) accurately generate requested digits and (2) be efficient in generating requested digits. Specifically, we sampled $10,000$ points from each mixture component in the latent representation, generated images, and used the model's internal clustering to assign a probability to which digits were generated. To generate correct and high-quality images with VaDE, the posterior of the latent representation needs to be evaluated for each sample. This was done for the different thresholds $\phi \in [0.0, 0.1, 0.2, \cdots, 0.9, 0.999]$. The default threshold Jiang et al. (2017) used was $\phi = 0.999$. Instead of thresholding the latent representation, we ran the generation process for MoE-Sim-VAE for each threshold with the same settings. To generate images from VaDE we used the Python implementation[1] and model weights publicly available from Jiang et al. (2017).

As a result of this analysis we report a confusion matrix for MoE-Sim-VAE in Figure A6, the confusion matrices for each threshold for VaDE in Figure A7, the accuracy of generating a requested digit and the number of runs required in Figure A5. In summary, one can see that the MoE-Sim-VAE generates digits more accurately with fewer resources required. This can especially be seen when comparing the number of iterations required to fulfill the default posterior threshold of $0.999$. VaDE needs nearly 2 million iterations to find samples that fulfill the aforementioned threshold criterion whereas the MoE-Sim-VAE only requires $10,000$ for a comparable sample accuracy. In comparison the mean accuracy over all thresholds for MoE-Sim-VAE is $0.970$, whereas VaDE reaches on average $0.944$. VaDE reaches a maximum accuracy of $0.995$, which costs the aforementioned 2 million iterations for generating $100,000$ images, whereas MoE-Sim-VAE reaches a maximum accuracy of $0.971$ with $100,000$ runs, without accounting for a systematic generating/clustering error (confusing 5 and 8) of MoE-Sim-VAE which can be seen in the confusion matrix in Figure A6.

### 4.3 LEARNING CELL TYPE COMPOSITION IN PERIPHERAL BLOOD MONONUCLEAR CELLS USING CYTOF MEASUREMENTS

In the following, we want to show representation learning performance on a real-world problem in biology. Specifically, we focus on cell type definition from single-cell measurements. Cytometry by time-of-flight mass spectrometry (CyTOF) (Bandura et al., 2009) is a state-of-the-art technique allowing measurement of up to $1,000$ cells per second and in parallel over $40$ protein markers of the cells (Kay et al., 2013). Defining biologically relevant cell subpopulations by clustering this data is a common learning task (Aghaeepour et al., 2013; Weber & Robinson, 2016).

Many methods have been developed to tackle the problem introduced above and were compared on four publicly available datasets in Weber & Robinson (2016). The best out of $18$ methods were FlowSOM (Gassen et al., 2015), PhenoGraph (Levine et al., 2015) and X-shift (Samusik et al., 2016). These are based on k-nearest-neighbors heuristics, either defined from a spanning graph or from estimating the data density. In contrast to these methods, MoE-Sim-VAE can map new cells into the latent representation, assign probabilities for cell types and infer an interpretable latent representation allowing intuitive downstream analysis by domain experts.

We applied MoE-Sim-VAE to the same datasets as in Weber & Robinson (2016) and achieve superior results in classification using the F-measure (Equation 12) in three out of four datasets. Similarly as in Weber & Robinson (2016) we trained MoE-Sim-VAE 30 times and report in Table 2 (adopted from Weber & Robinson (2016)) the means across all runs. The reproducibility of our model for each dataset can be seen in Figure A8.

Further, we trained a MoE-Sim-VAE model on 268 datasets from Bodenmiller et al. (2012) (more details on the data in A.1.3), and achieve superior classification results of cell subpopulations in the data when comparing to state-of-the-art methods in this field (PhenoGraph, X-Shift, FlowSOM).

---

[1] https://github.com/slim1017/VaDE

Table 2: Comparison of MoE-Sim-VAE performance to competitor methods in defining cell type composition in CyTOF measurements. The results in the table are extracted from the review paper of Weber & Robinson (2016), where 18 methods are compared on four different datasets. Our model outperforms the baselines on four out of five data sets.

| Method | Levine_32dim | Levine_13dim | Samusik_01 | Samusik_all |
|---|---|---|---|---|
| ACCENSE | 0.494 | 0.358 | 0.517 | 0.502 |
| ClusterX | 0.682 | 0.474 | 0.571 | 0.603 |
| DensVM | 0.66 | 0.448 | 0.239 | 0.496 |
| FLOCK | 0.727 | 0.379 | 0.608 | 0.631 |
| flowClust | NA | 0.416 | 0.612 | 0.61 |
| flowMeans | 0.769 | 0.518 | 0.625 | 0.653 |
| flowMerge | NA | 0.247 | 0.452 | 0.341 |
| flowPeaks | 0.237 | 0.215 | 0.058 | 0.323 |
| FlowSOM | **0.78** | 0.495 | 0.707 | 0.702 |
| FlowSOM_pre | 0.502 | 0.422 | 0.583 | 0.528 |
| immunoClust | 0.413 | 0.308 | 0.552 | 0.523 |
| kmeans | 0.42 | 0.435 | 0.65 | 0.59 |
| PhenoGraph | 0.563 | 0.468 | 0.671 | 0.653 |
| Rclusterpp | 0.605 | 0.465 | 0.637 | 0.613 |
| SamSPECTRAL | 0.512 | 0.253 | 0.263 | 0.138 |
| SPADE | NA | 0.127 | 0.169 | 0.13 |
| SWIFT | 0.177 | 0.179 | 0.202 | 0.208 |
| Xshift | 0.691 | 0.47 | 0.679 | 0.657 |
| *MoE-Sim-VAE* (proposed) | 0.70 | **0.68** | **0.76** | **0.74** |

Exact results can be seen in Table A1 or visualized in Figure 3. More details on the MoE-Sim-VAE setting used for all results on CyTOF data are reported in the appendix (Section A.1.3).

## 5 CONCLUSION

Our MoE-Sim-VAE model can infer similarity-based representations, perform clustering tasks, and efficiently as well as accurately generate high-dimensional data. The training of the model is performed by optimizing a joint objective function consisting of data reconstruction, clustering, and KL loss, where the latter regularizes the latent representation. On synthetic data, we have shown the strengths and limitations of the model. On the benchmark dataset of MNIST, we presented superior clustering performance and the efficiency and accuracy of MoE-Sim-VAE in generating high-dimensional data. On the biological real-world task of defining cell subpopulations in complex single-cell data, we show superior clustering performances compared to state-of-the-art methods on over 270 datasets and therefore demonstrate MoE-Sim-VAE's real-world usefulness.

Future work might include to add adversarial training to the MoE decoder, which could improve image generation to create even more realistic images. Also, specific applications might benefit from replacing the Gaussian with a different mixture model. So far the MoE-Sim-VAE's similarity measure has to be defined by the user. Relaxing this requirement and allowing for learning a useful similarity measure automatically for inferring latent representations will be an interesting extension to explore. This could be useful in a weakly-supervised setting, which often occurs for example in clinical data consisting of healthy and diseased patients. Minor details between a healthy and diseased patient might make a huge difference and could be learned from the data using neural networks.

## REFERENCES

N. Aghaeepour, G. Finak, FlowCAP Consortium, DREAM Consortium, H. Hoos, TR. Mosmann, R. Brinkman, R. Gottardo, and Rh. Scheuermann. Critical assessment of automated flow cytometry data analysis techniques. *Nature Methods*, 2013.

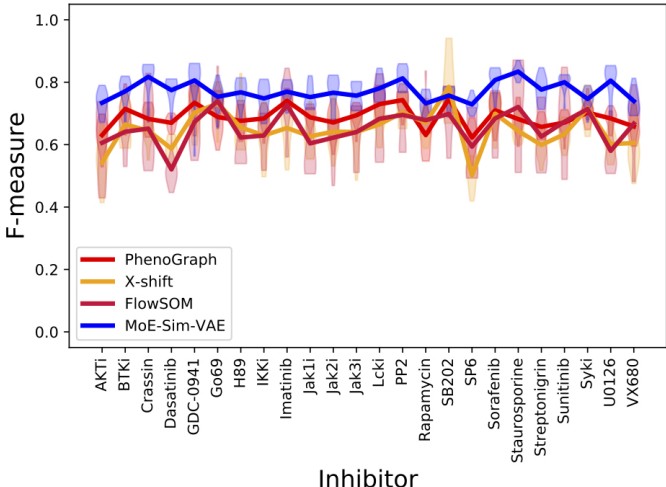

Figure 3: Comparison of MoE-Sim-VAE to the most popular competitor methods on defining cell types in peripheral blood mononuclear cell data via CyTOF measurements. On the x-axis different inhibitor treatments are listed whereas the y-axis reports the respective F-measure, defined in Equation 12, as performance measure of the methods. Each violin plot represents a run on a different inhibitor with multiple wells, whereas the line connects the means of the performance on the specific inhibitor.

Elie Aljalbout, Vladimir Golkov, Yawar Siddiqui, Maximilian Strobel, and Daniel Cremers. Clustering with deep learning: Taxonomy and new methods. *arXiv*, 2018.

DR. Bandura, VI. Baranov, OI. Ornatsky, A. Antonov, R. Kinach, X. Lou, S. Pavlov, S. Vorobiev, JE. Dick, and SD. Tanner. Mass cytometry: Technique for real time single cell multitarget immunoassay based on inductively coupled plasma time-of-flight mass spectrometry. *Analytical Chemistry*, 2009.

Christopher M. Bishop. *Neural Networks for Pattern Recognition*. CLARENDON PRESS, 1995.

Christopher M. Bishop. *Pattern Recognition and Machine Learning*. Springer, 2006.

Bernd Bodenmiller, Eli R. Zunder, Rachel Finck, Tiffany J. Chen, Erica S. Savig, Robert V. Bruggner, Erin F. Simonds, Sean C. Bendall, Peter O. Krutzik Karen Sachs, and Garry P. Nolan. Multiplexed mass cytometry profiling of cellular states perturbed by small-molecule regulators. *Nature Biotechnology*, 2012.

D. Chen, J. Lv, and Z. Yi. Unsupervised multi-manifold clustering by learning deep representation. *Workshops at the AAAI Conference on Artificial Intelligence*, 2017.

S. Chopra, R. Hadsell, and Y. LeCun. Learning a similarity metric discriminatively, with application to face verification. *IEEE*, 2005.

Kamran Ghasedi Dizaji, Amirhossein Herandi, Cheng Deng, Weidong Cai, and Heng Huang. Deep clustering via joint convolutional autoencoder embedding and relative entropy minimization. *arXiv*, 2017.

Vincent Fortuin, Matthias Hüser, Francesco Locatello, Heiko Strathmann, and Gunnar Rätsch. Somvae: Interpretable discrete representation learning on time series. *Conference paper at ICLR 2019*, 2019.

Sofie Van Gassen, Britt Callebaut, Mary J. Van Helden, Bart N. Lambrecht, Piet Demeester, Tom Dhaene, and Yvan Saeys. Flowsom: Using self-organizing maps for visualization and interpretation of cytometry data. *Cytometry Part A*, 2015.

Arthur Gretton, Karsten Borgwardt, Malte J. Rasch, Bernhard Scholkopf, and Alexander J. Smola. A kernel method for the two-sample problem. *arXiv*, 2008.

C.-C. Hsu and C.-W. Lin. Cnn-based joint clustering and representation learning with feature drift compensation for large-scale image data. *arXiv*, 2017.

W. Hu, T. Miyato, S. Tokui, E. Matsumoto, and M. Sugiyama. Learning discrete representations via information maximizing self augmented training. *arXiv*, 2017.

Jasmine Irani, Nitin Pise, and Madhura Phatak. Clustering techniques and the similarity measures used in clustering: A survey. *International Journal of Computer Applications*, 2016.

Zhuxi Jiang, Yin Zheng, Huachun Tan, Bangsheng Tang, and Hanning Zhou. Variational deep embedding: An unsupervised and generative approach to clustering. *arXiv*, 2017.

Alexander W. Kay, Dara M. Strauss-Albee, and Catherine A. Blish. Application of mass cytometry (cytof) for functional and phenotypic analysis of natural killer cells. *Methods in Molecular Biology*, 2013.

D. P. Kingma and M. Welling. Auto-encoding variational bayes. *International Conference on Learning Representations (ICLR)*, 2014.

Yann LeCun, Leon Botto, Yoshua Bengi, and Patrick Haffner. Gradient-based learning applied to document recognition. *Proceedings ofthe IEEE*, 1998.

Jacob H. Levine, Erin F. Simonds, Sean C. Bendall, Kara L. Davis, El ad D. Amir, Michelle D. Tadmor, Oren Litvin, Harris G. Fienberg, Astraea Jager, Eli R. Zunder, Rachel Finck, Amanda L. Gedman, Ina Radtke, James R. Downing, Dana Peer, and Garry P. Nolan. Data-driven phenotypic dissection of aml reveals progenitor-like cells that correlate with prognosis. *Cell*, 2015.

F. Li, H. Qiao, B. Zhang, and X. Xi. Discriminatively boosted image clustering with fully convolutional autoencoders. *arXiv*, 2017.

Leland McInnes, John Healy, and James Melville. Umap: Uniform manifold approximation and projection for dimension reduction. *arXiv*, 2018.

Erxue Min, Xifeng Guo, Qiang Liu, Gen Zhang, Jianjing Cui, and Jun Long. A survey of clustering with deep learning: From the perspective of network architecture. *IEEE*, 2018.

Peng Qiu, Erin F. Simonds, Sean C. Bendall, Kenneth D. Gibbs Jr., Robert V. Bruggner, Michael D. Linderman, Karen Sachs, Garry P. Nolan, and Sylvia K. Plevritis. Extracting a cellular hierachy from high-dimensional cytometry data with spade. *Nature Biotechnology*, 2011.

S. Saito and R. T. Tan. Neural clustering: Concatenating layers for better projections. *Workshop track at ICLR 2017*, 2017.

Nikolay Samusik, Zinaida Good, Matthew H. Spitzer, Kara L. Davis, and Garry P. Nolan. Automated mapping of phenotype space with single-cell data. *Nature Methods*, 2016.

Uri Shaham, Kelly Stanton, Henry Li, Boaz Nadler, Ronen Basri, and Yuval Kluger. Spectralnet: Spectral clustering using deep neural networks. *Published as a conference paper at ICLR 2018*, 2018.

Noam Shazeer, Azalia Mirhoseini, Krzysztof Maziarz, Andy Davis, Quoc Le1, Geoffrey Hinton, and Jeff Dean. Outrageously large neural networks: The sparsely-gated mixture-of-experts layers. *arXiv*, 2017.

Dougal J. Sutherland, Hsiao-Yu Tung, Heiko Strathmann, Soumyajit De, Aaditya Ramdas, Alex Smola, and Arthur Gretton. Generative models and model criticism via optimized maximum mean discrepancy. *arXiv*, 2019.

Laurens van der Maaten and Geoffrey Hinton. Visualizing data using t-sne. *Journal of Machine Learning Research*, 2008.

Z. Wang, S. Chang, J. Zhou, M. Wang, and T. S. Huang. Learning a task-specific deep architecture for clustering. *Proceedings of the SIAM International Conference on Data Mining (ICDM)*, 2016.

Lukas M. Weber and Mark D. Robinson. Comparison of clustering methods for highdimensional singlecell flow and mass cytometry data. *Cytometry Part A*, 2016.

J. Xie, R. Girshick, and A. Farhadi. Unsupervised deep embedding for clustering analysis. *International Conference on Machine Learning (ICML)*, 2016.

Bo Yang, Xiao Fu, Nicholas D. Sidiropoulos, and Mingyi Hong. Towards k-means-friendly spaces: Simultaneous deep learning and clustering. *arXiv*, 2017.

J. Yang, D. Parikh, and D.Batra. Joint unsupervised learning of deep representations and image clusters. *Proceedings of the IEEE Conference on Computer Vision and Pattern Recognition (CVPR)*, 2016b.

Dejiao Zhang, Yifan Sun, Brian Eriksson, and Laura Balzano. Deep unsupervised clustering using mixture of autoencoders. *arXiv*, 2017.

## A    APPENDIX

### A.1    EXPERIMENTAL DETAILS

In the following sections we provide more details on model implementations, metrics used and additional result figures for the experiments described in the main text.

#### A.1.1    EVALUATION OF MOE-SIM-VAE ON SYNTHETIC DATA

Model and training details:

- number of experts: $\{2, \ldots, 40\}$
- batch size: 512
- code size: 10
- Number of iterations: 5000
- activation function; elu
- loss coefficient data reconstruction: 0.487
- loss coefficient clustering : 0.487
- loss coefficient mixture of Gaussian: 0.024
- learning rate: 0.001
- batch normalization
- dropout rate: 0.5
- distance threshold (perplexity parameter): 2
- depth clustering network: 5
- internal size clustering network: 100
- trainable parameters: depending on number of experts

We compare results based on F-measure (Aghaeepour et al., 2013), which is defined as follows

$$F(C, K) = \sum_{c_i \in C} \frac{|c_i|}{N} \max_{k_j \in K} \{F(c_i, k_j)\} \tag{12}$$

where $N$ is the number of samples $C\{c_1, c_2, \ldots, c_n\}$ and $K\{k_1, k_2, \ldots, k_m\}$ are the cluster result and the reference cluster, respectively. Further $F(c_i, k_j)$ is the harmonic mean of precision and recall according to

$$F(c_i, k_j) = \frac{2Pr(c_i, k_j)Re(c_i, k_j)}{Pr(c_i, k_j) + Re(c_i, k_j)} \tag{13}$$

whereby $Pr(c_i, k_j)$ is the precision and $Re(c_i, k_j)$ is the recall. Results are shown in Table 2.

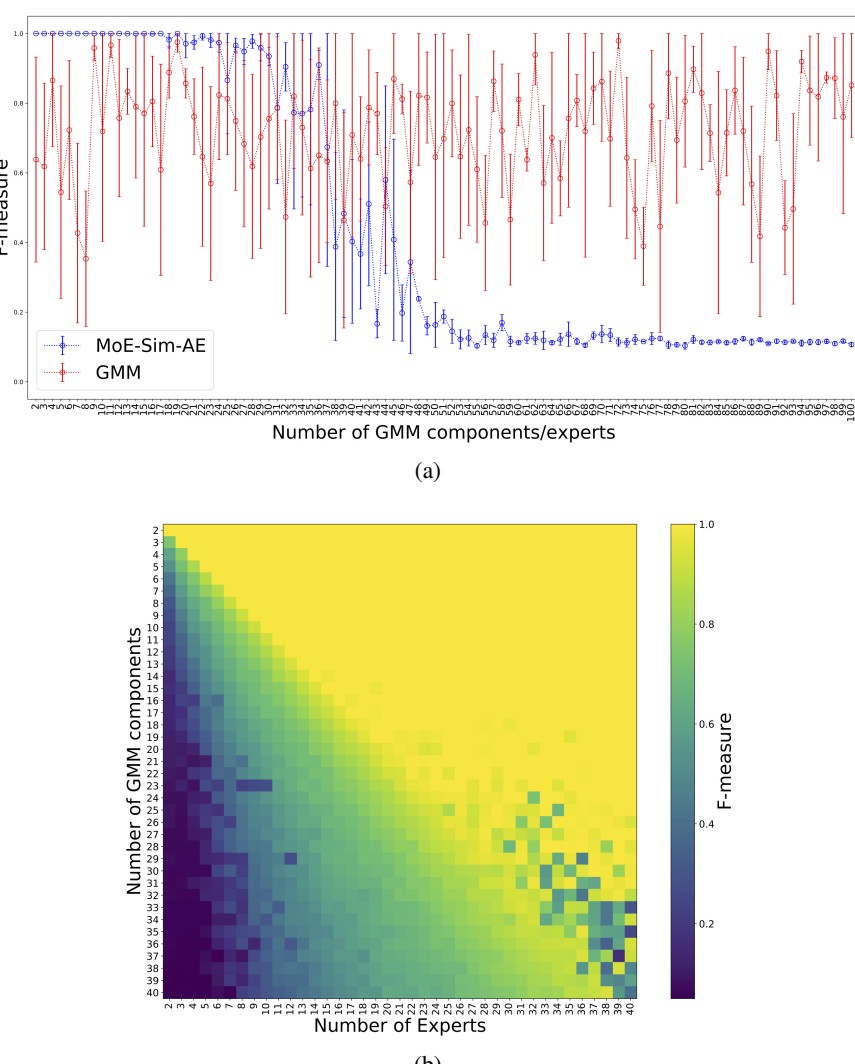

(a)

(b)

Figure A1: Testing MoE-Sim-VAE on data sampled from a Gaussian mixture model with random sampled parameters. Figure A1a: Testing with exact numbers of experts. When comparing to classification results with GMMs one can see that our model achievs better results until around 30 mixture components and is still compatitive until 40 mixture components. With more then 40 mixture components ourr MoE-Sim-VAE is not able anymore to compete with a GMM. Figure A1b: Testing for specific number of synthetic mixture components and iterating number of experts. Until a number of GMM components of 23 MoE-Sim-VAE is very precise in learning the real number of clusters even when allowing the model to have 40 experts.

### A.1.2 UNSUPERVISED CLUSTERING, EMBEDDING AND DATA GENERATION OF MNIST

Model and training details:

- number of experts: 10
- batch size: 128
- code size: 68
- Number of iterations: 20000
- activation function; elu
- loss coefficient data reconstruction: 0.487

- loss coefficient clustering : 0.487
- loss coefficient mixture of Gaussian: 0.024
- learning rate: 0.0001
- batch normalization
- dropout rate: 0.5
- k from kNN (perplexity parameter): 10
- depth clustering network: 3
- internal size clustering network: 200
- trainable parameters: 1619446

One estimator of the Maximum Mean Discrepancy (MMD) (Gretton et al., 2008) is defined as

$$\widehat{MMD}^2(X,Y) = \frac{1}{\binom{m}{2}} \sum_{i \neq i'} k(X_i, X_{i'}) + \frac{1}{\binom{m}{2}} \sum_{j \neq j'} k(Y_j, Y_{j'}) - \frac{2}{\binom{m}{2}} \sum_{i,j} k(X_i, Y_j) \qquad (14)$$

where $X = \{\hat{x}_1, \cdots, \hat{x}_m\} \overset{iid}{\sim} P, Y = \{\hat{y}_1, \cdots, \hat{y}_m\} \overset{iid}{\sim} Q$ are samples from two distributions (e.g. samples from two different clusters of the latent representation, for MNIST of two different digits) and $k$ is a kernel function, where we use the popular RBF kernel. Based on that estimator Sutherland et al. (2019) introduced the hypothesis test

$$H0 : P = Q \qquad (15)$$
$$H1 : P \neq Q \qquad (16)$$

using the statistic $m\widehat{MMD}^2(X,Y)$. The distribution for $P$ and $Q$ is not required to be known. Sutherland et al. (2019) used MMD and this test to train a Generative Adversarial Network (GAN) and also to evaluate the generative performance of the model. In this work we use $\widehat{MMD}^2(X,Y)$ to test if samples of different clusters of the latent representation are similar, or in other words the distance of the distributions. We used the Python implementation[2] from Sutherland et al. (2019).

### A.1.3 LEARNING CELL TYPE COMPOSITION IN PERIPHERAL BLOOD MONONUCLEAR CELLS USING CYTOF MEASUREMENTS

Model and training details for all experiments on CyTOF data:

- number of experts: 25 (Weber & Robinson, 2016), 15 (Bodenmiller et al., 2012)
- batch size: 128
- code size: 9
- Number of iterations: 30000 (Weber & Robinson, 2016), 20000 (Bodenmiller et al., 2012)
- activation function: relu
- loss coefficient data reconstruction: 1
- loss coefficient clustering : 1
- loss coefficient mixture of Gaussian: 0
- learning rate: 0.001 (Weber & Robinson, 2016), 0.005 (Bodenmiller et al., 2012)
- batch normalization
- dropout rate: 0.5
- distance threshold (perplexity parameter): 2
- distance metric: correlation
- depth clustering network: 5
- internal size clustering network: 9

---

[2]`https://github.com/dougalsutherland/opt-mmd/blob/master/two_sample/mmd_test.py`

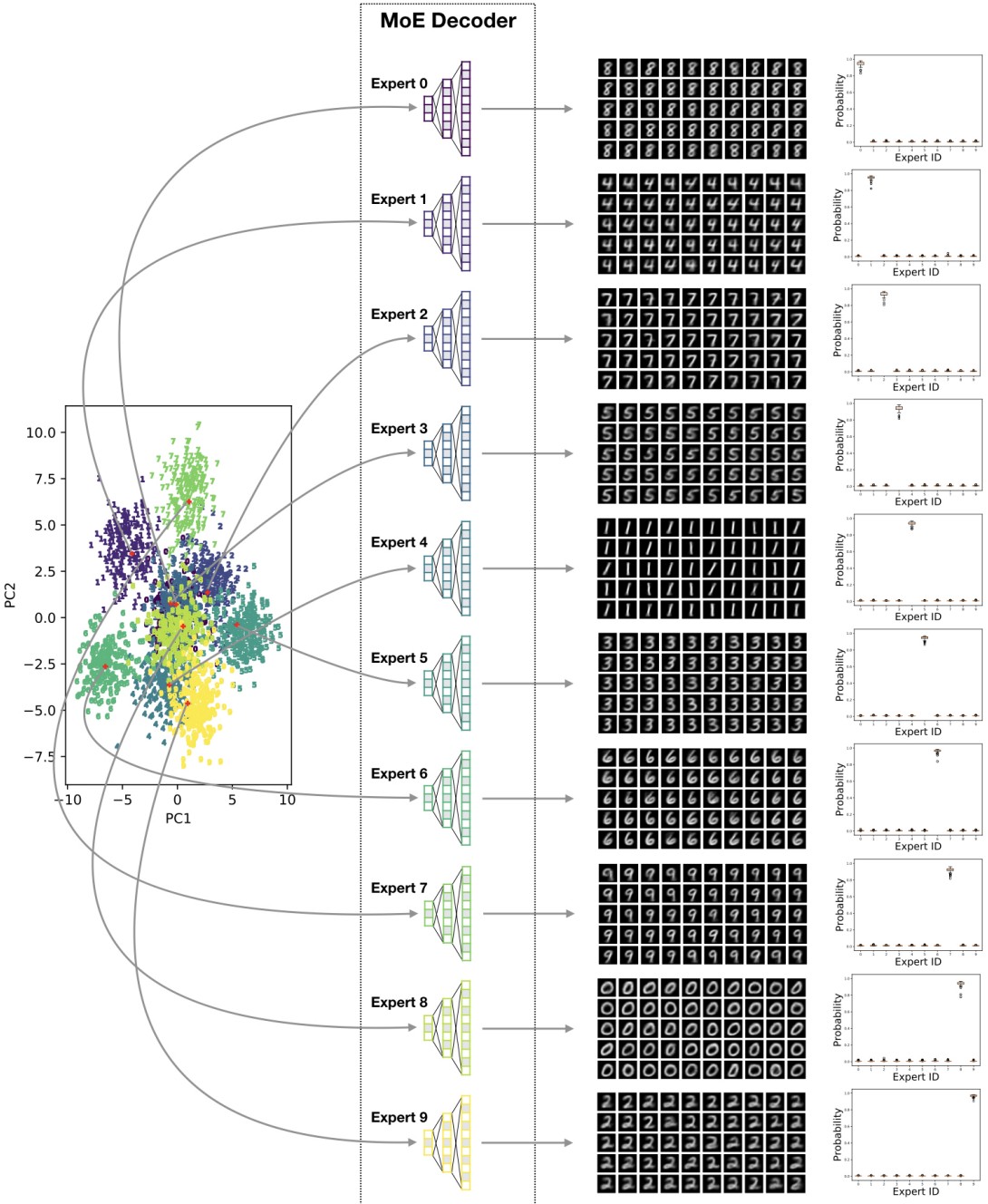

Figure A2: More detailed overview of results and generated samples of MNIST images. The plot on the left side shows the latent representation where the red crosses are the cluster centers. Those can be used as a mean to sample from a standard Gaussian for data generation via the MoE Decoder. The boxplots on the right show the clustering and gating result of each sample from the variational distribution.

- trainable parameters: 37563 (Weber & Robinson, 2016), 22228 (Bodenmiller et al., 2012)

Results are computed setting the loss coefficient for the KL loss 3 equal to zero, since we do not intend to generate any data, but rather give the chance to the AE to pick up the correct subpouplations.

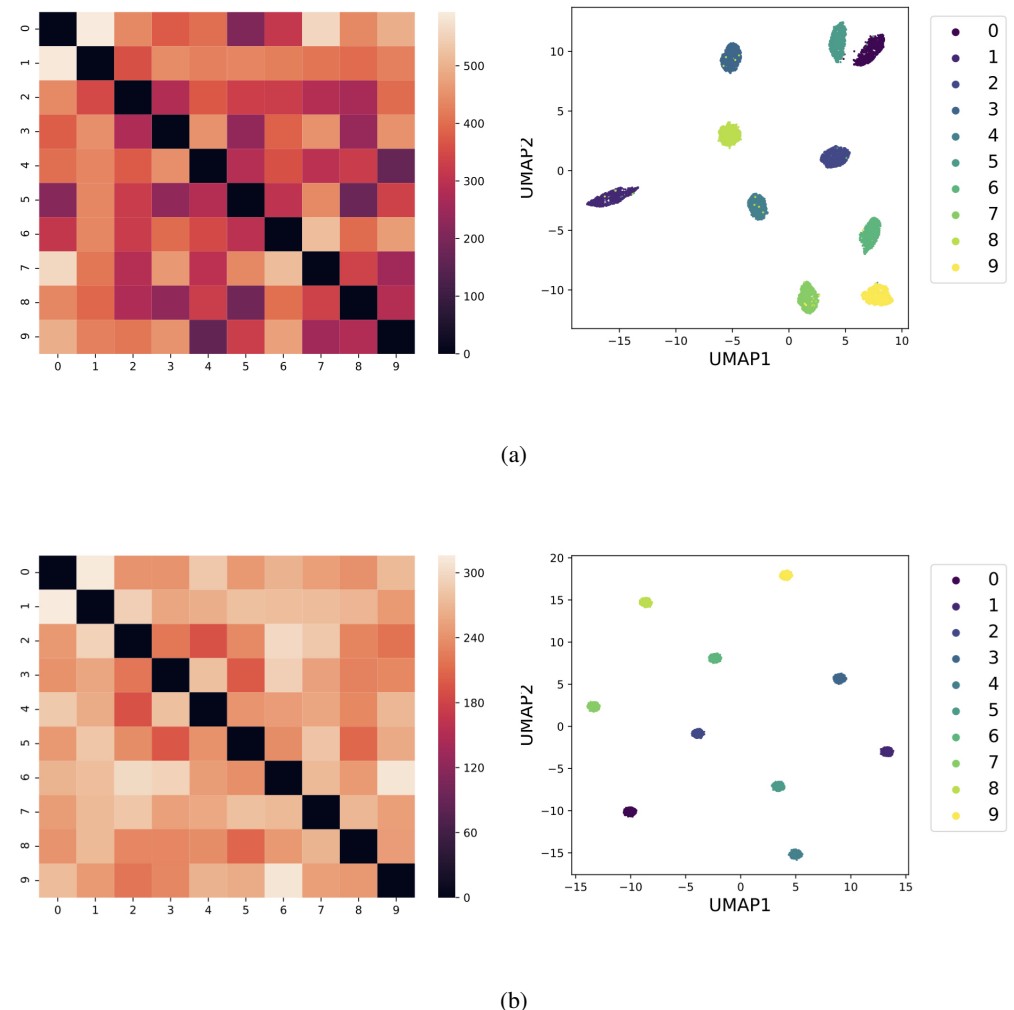

(a)

(b)

Figure A3: Comparison of two sample MMD test (Sutherland et al., 2019) on the distributions from the different mixture components in the latent representation. The heatmaps on the left side show the estimation of the MMD which can be seen as the distance between pairs of distributions. The figures on the right side show the separation of the cluster in the latent representation based on a dimensionality reduction via UMAP (McInnes et al., 2018). Figure A3a shows the results for the clusters of VaDE at a posterior threshold of $0.8$ which is the first threshold which shows total separation of all clusters. Figure A3b shows the separation of the clusters in latent space learned from MoE-Sim-VAE. For both methods, all distributions belonging to clusters of different respective digits show a larger distance compared to the diagonal of matching distributions, such that we generate images from a well-separated latent representation for both methods and therefore the main difference comes from the decoders.

Also here we use the F-measure defined in Equation 12 as metric to evaluate the models. For the data compared in Weber & Robinson (2016) we ran each model 30 times and report reproducability of our results in A8. The model was trained on all data and validated on the on with labels.

For the data from Bodenmiller et al. (2012) we run each model on one time on the each of the 268 datasets. Hereby we focused on the following surface markers: CD3(110:114)Dd, CD45(In115)Dd, CD4(Nd145)Dd, CD20(Sm147)Dd, CD33(Nd148)Dd, CD123(Eu151)Dd, CD14(Gd160)Dd, IgM(Yb171)Dd, HLA-DR(Yb174)Dd, CD7(Yb176)Dd. The subpopulations were originally defined via the SPADE algorithm (Qiu et al., 2011), which is a visualization tool using Agglomerative

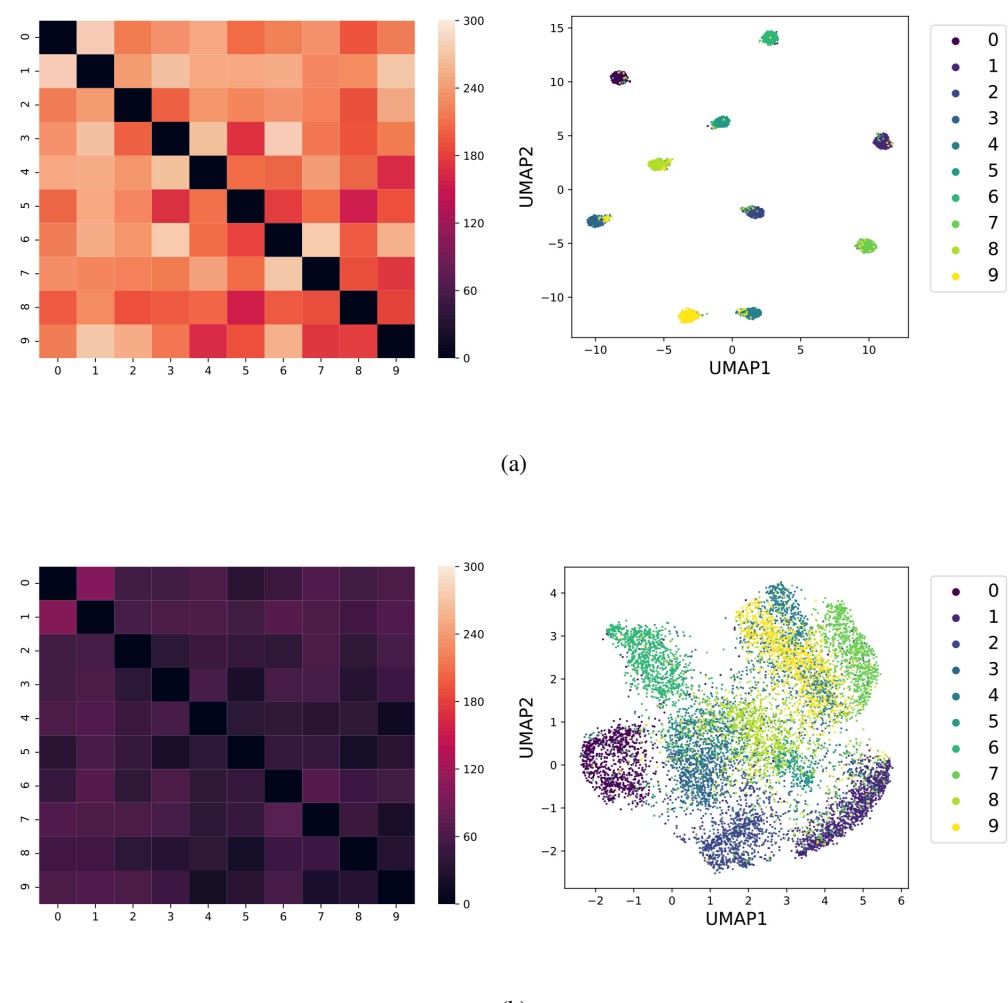

(a)

(b)

Figure A4: Ablation study on the similarity matrix $S$. Both figures show the MMD statistic and UMAP (McInnes et al., 2018) projection of reconstructed MNIST digits computed on the latent representation. Figure A4a shows the results on MoE-Sim-VAE trained with the similarity matrix. The different digits separate well which can also be seen in the heatmap showing the MMD statistics between all digits. In comparison, Figure A4b shows results of the MoE-Sim-VAE model ignoring the similarity matrix setting the loss coefficient to zero. One can observe that the MMD statistic, which can be seen as a similarity measure of two distributions, is way lower compared to the model including the similarity matrix. Further, also the UMAP projection confirms less separation in the latent representation between the different digits.

hierarchical clustering and minimum spanning trees. The gating of the cells is done manually via coloring of the tree leaves. With MoE-Sim-VAE we try to reconstruct the defined manually defined subpopulations. Bodenmiller et al. (2012) performed experiments on multiple well plates were different inhibitors and their effect was tested. We selected for each well plate row A to test our model on. We decided for all methods to discard subpopulations which are smaller then 30 cells. As a similarity measure for MoE-Sim-VAE we reduced the dimension of the data using UMAP (McInnes et al., 2018) using the Canberra distance

$$d(\boldsymbol{p}, \boldsymbol{q}) = \sum_{i=1}^{n} \frac{|p_i - q_i|}{|p_i| + |q_i|} \tag{17}$$

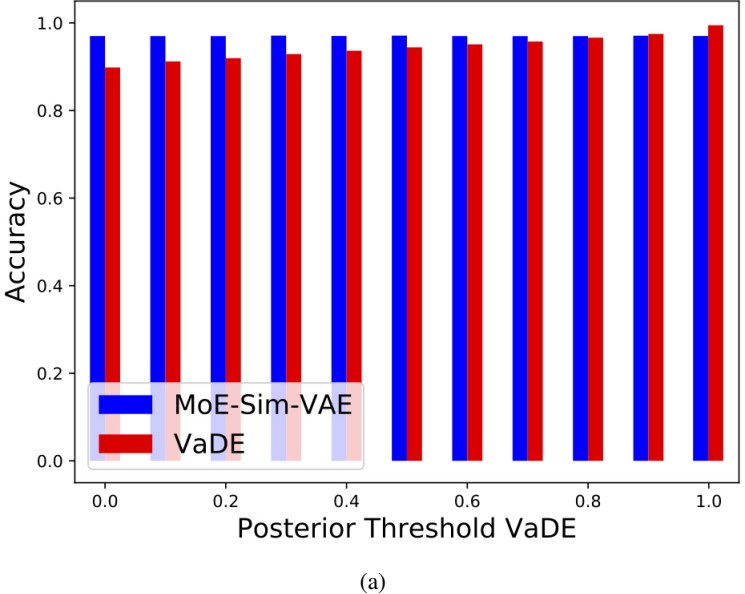

(a)

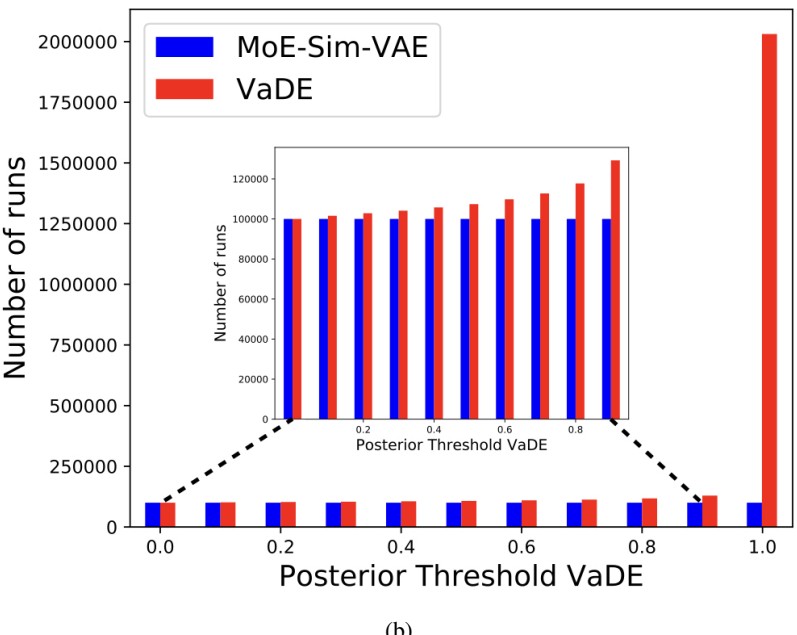

(b)

Figure A5: Comparison of data generation process between Moe-Sim-VAE and VaDE (Jiang et al., 2017). Figure A5a shows the accuracy of how certain a specific digit can be generated from the respective cluster in the latent representation whereas Figure A5b compares the number of runs until a sample from the latent representation satisfied the posterior criterion from VaDE. It needs to be mentioned that MoE-Sim-VAE does not require any thresholding such that we ran the data generation process multiple times with the same settings to compare with VaDE. In total $10000$ samples are generated for each digit.

where $\boldsymbol{p} = (p_1, p_2, \ldots, p_n)$ and $\boldsymbol{q} = (q_1, q_2, \ldots, q_n)$. Cells were defined to be similar in MoE-Sim-VAE when the distance between the cells in the UMAP-projection was smaller then a threshold. We trained and tested MoE-Sim-VAE on a splitted dataset with rations $0.8/0.2$ and evaluated the performance on the unseen test dataset. In comparison the compatitor methods were trained and

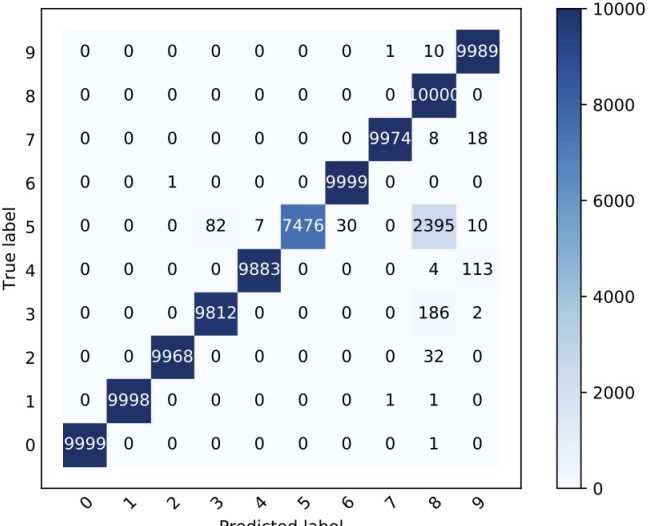

Figure A6: Confusion map for data generation using MoE-Sim-VAE. Besides the systematic error of confusing digit 5 and 8, which can also depend on the clustering network, the digit generation of our model performs very precise with a high accuracy of generating the digit asked for. In comparison to VaDE (Jiang et al., 2017) our model does not need any threshold on samples from the latent representation which reduces the computational costs by far.

tested on all the data, which is an advantage in comparison to our model, but still MoE-Sim-VAE outpreforms the compatitors.

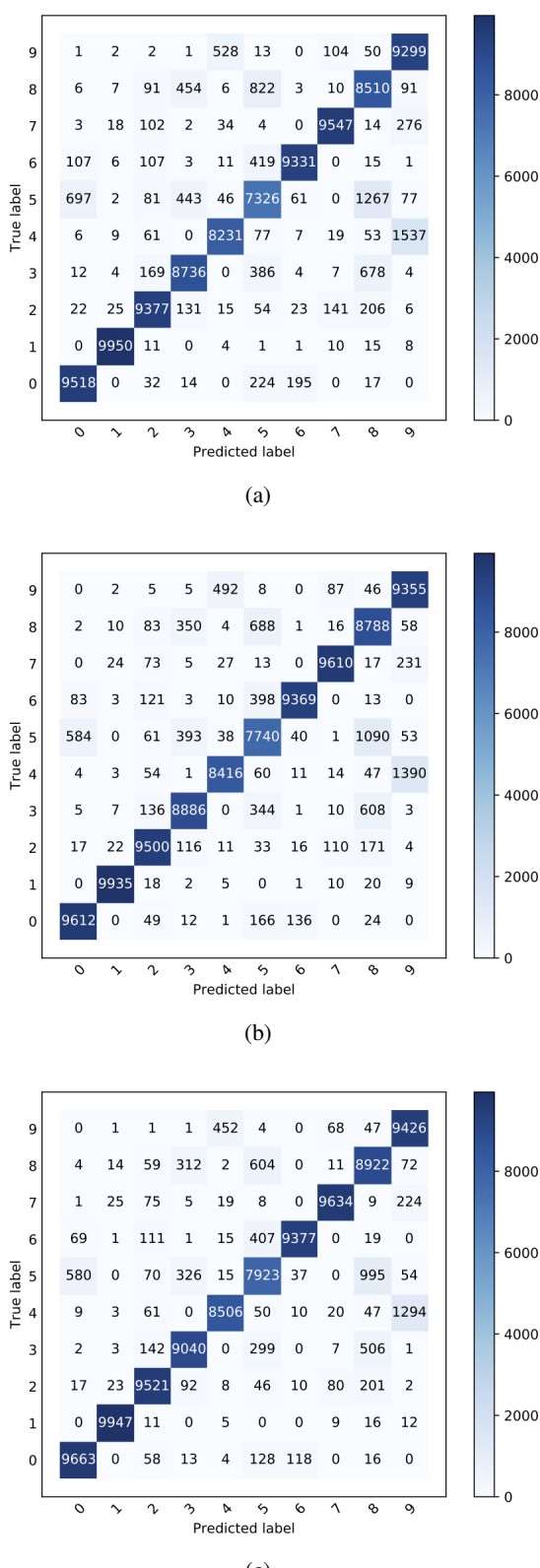

Figure A7: Confusion maps for data generation using VaDE.
Figure A7a Posterior threshold $0.0$.
Figure A7b Posterior threshold $0.1$.
Figure A7c Posterior threshold $0.2$.

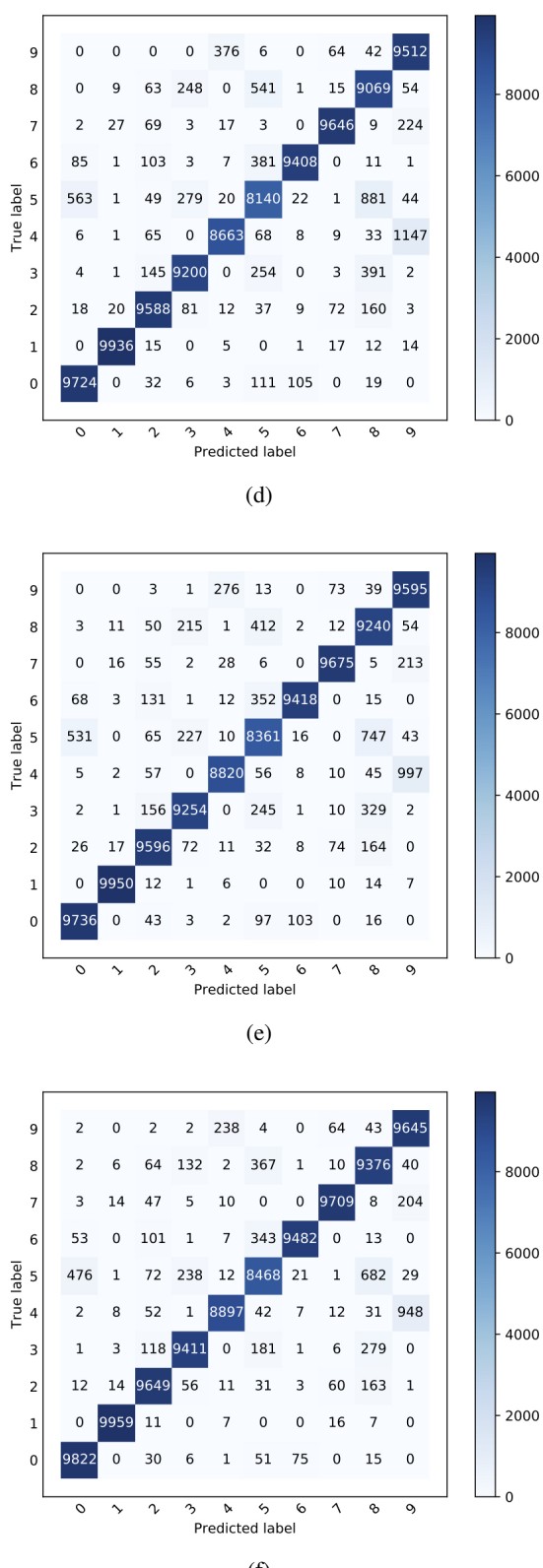

Figure A7: Confusion maps for data generation using VaDE.
Figure A7d Posterior threshold $0.3$.
Figure A7e Posterior threshold $0.4$.
Figure A7f Posterior threshold $0.5$.

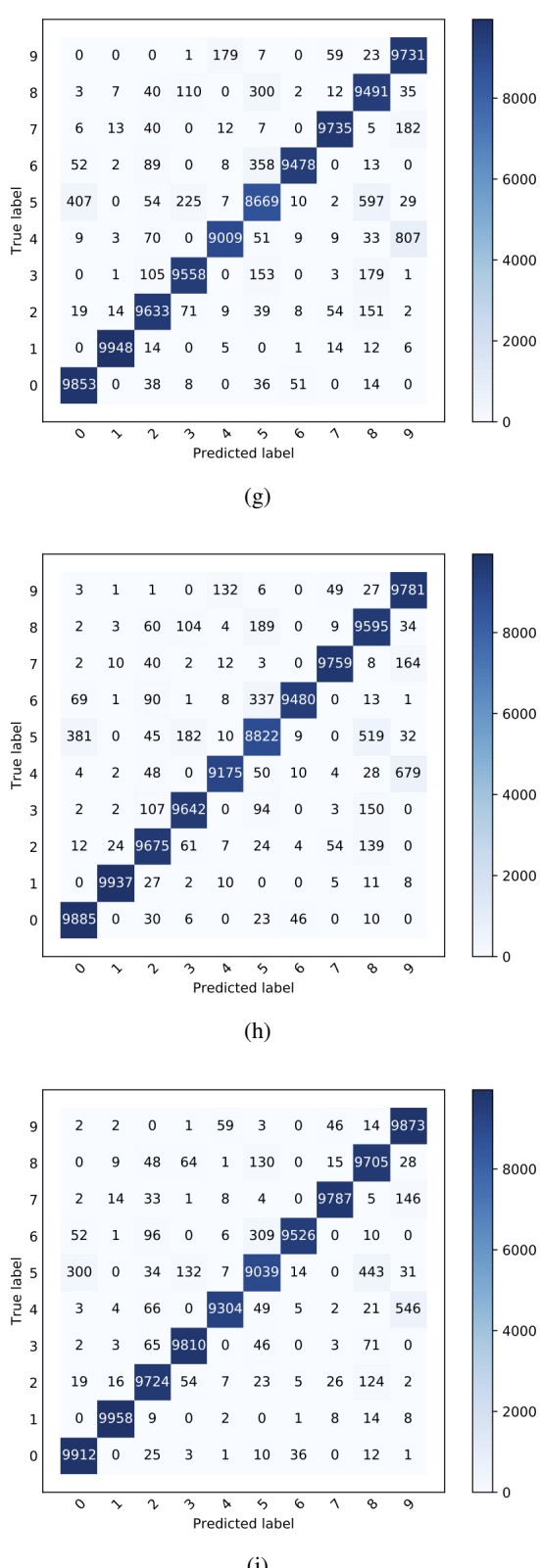

Figure A7: Confusion maps for data generation using VaDE.
Figure A7g Posterior threshold $0.6$.
Figure A7h Posterior threshold $0.7$.
Figure A7i Posterior threshold $0.8$.

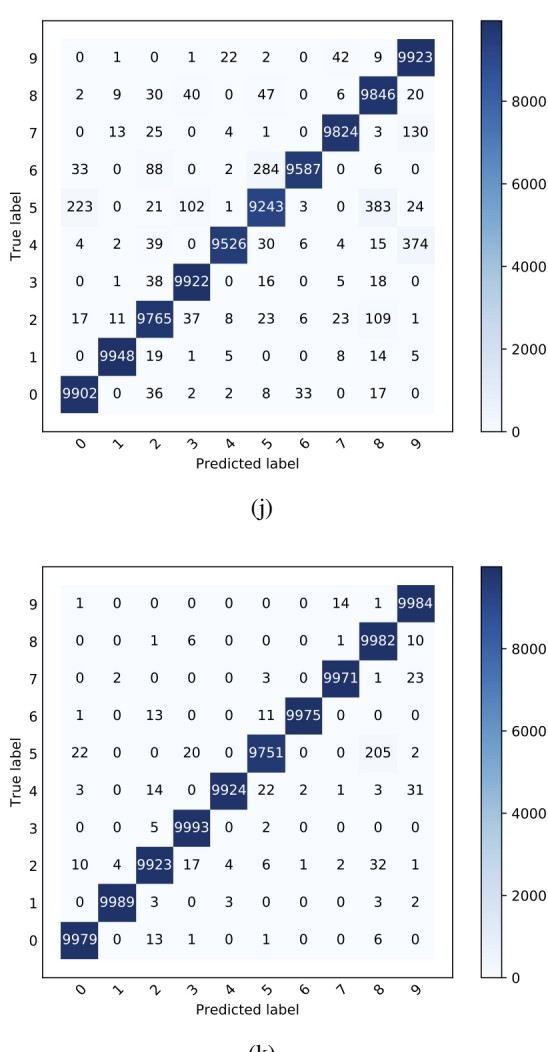

(j)

(k)

Figure A7: Confusion maps for data generation using VaDE.
Figure A7j Posterior threshold 0.9.
Figure A7k Posterior threshold 0.999. (default for Jiang et al. (2017))

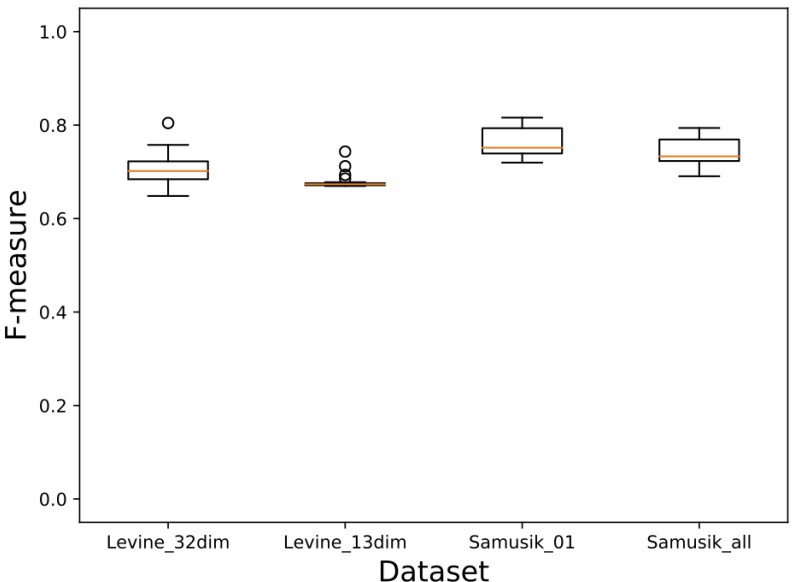

Figure A8: The boxplots show similar as in Weber & Robinson (2016) the reproducibility of MoE-Sim-VAE on the four datasets when running MoE-Sim-VAE 30 times. The variance on defining the correct subpopulations of MoE-Sim-VAE is quite small and therefore also an improvment to many methods compared in Weber & Robinson (2016).

Table A1: Results of MoE-Sim-VAE on data published in Bodenmiller et al. (2012). CyTOF measurements from peripheral blood mononuclear cells (PBMCs) were taken and the goal is to define the different cell types present in the data. The ground truth was definied using the SPADE algorithm (Qiu et al., 2011), which can visualize the high dimensional data in such a way to be able to manual gate the cells. We compare to other fully unsupervised methods as FlowSOM, X-shift and PhenoGraph and achieve in most cases the best F-measure, which is defined as in Equation 12.

| Inhibitor | Well | MoE-Sim-VAE | FlowSOM | X-shift | PhenoGraph |
|-----------|------|-------------|---------|---------|------------|
| AKTi | A02 | **0.7666** | 0.5147 | 0.5704 | 0.6588 |
| AKTi | A03 | **0.7541** | 0.4793 | 0.546 | 0.6026 |
| AKTi | A04 | **0.6815** | 0.6405 | 0.5298 | 0.5974 |
| AKTi | A05 | **0.7127** | 0.7108 | 0.6089 | 0.6104 |
| AKTi | A06 | 0.6711 | **0.7383** | 0.572 | 0.6611 |
| AKTi | A07 | **0.7233** | 0.7034 | 0.5583 | 0.6981 |
| AKTi | A08 | **0.7901** | 0.7024 | 0.4541 | 0.5287 |
| AKTi | A09 | **0.7604** | 0.4292 | 0.5014 | 0.6414 |
| AKTi | A10 | **0.7275** | 0.4952 | 0.4144 | 0.677 |
| AKTi | A11 | **0.7540** | 0.6456 | 0.6673 | 0.6302 |
| BTKi | A02 | 0.7261 | 0.698 | 0.7136 | **0.7478** |
| BTKi | A03 | **0.7982** | 0.6643 | 0.6012 | 0.7141 |
| BTKi | A04 | **0.7835** | 0.6864 | 0.6983 | 0.7103 |
| BTKi | A05 | **0.7484** | 0.6397 | 0.7454 | 0.7474 |
| BTKi | A06 | **0.8196** | 0.703 | 0.7625 | 0.7949 |
| BTKi | A07 | **0.7976** | 0.6729 | 0.6841 | 0.7102 |
| BTKi | A08 | **0.8108** | 0.6715 | 0.5887 | 0.6884 |
| BTKi | A09 | **0.7789** | 0.5299 | 0.6426 | 0.7236 |
| BTKi | A10 | **0.7726** | 0.6319 | 0.6775 | 0.7148 |
| BTKi | A11 | **0.7857** | 0.6078 | 0.5939 | 0.6786 |
| BTKi | A12 | **0.6600** | 0.5503 | 0.6028 | 0.6308 |
| Crassin | A01 | **0.6727** | 0.6488 | 0.6315 | 0.6237 |
| Crassin | A02 | **0.8225** | 0.557 | 0.6435 | 0.7165 |
| Crassin | A03 | **0.8346** | 0.5736 | 0.6628 | 0.7085 |
| Crassin | A04 | **0.8446** | 0.5348 | 0.7146 | 0.7045 |
| Crassin | A05 | **0.8462** | 0.7444 | 0.6227 | 0.7202 |
| Crassin | A06 | **0.8569** | 0.7448 | 0.7078 | 0.6972 |
| Crassin | A07 | **0.8170** | 0.5164 | 0.6546 | 0.6309 |
| Crassin | A08 | **0.8431** | 0.8283 | 0.5504 | 0.6546 |
| Crassin | A09 | **0.8412** | 0.5814 | 0.6027 | 0.6684 |
| Crassin | A10 | **0.8527** | 0.7537 | 0.6586 | 0.6338 |
| Crassin | A11 | **0.8453** | 0.7174 | 0.6437 | 0.7358 |
| Crassin | A12 | **0.7320** | 0.6161 | 0.6436 | 0.6949 |
| Dasatinib | A01 | **0.7235** | 0.4466 | 0.554 | 0.6725 |
| Dasatinib | A02 | **0.8019** | 0.516 | 0.6238 | 0.701 |
| Dasatinib | A03 | **0.7864** | 0.5108 | 0.5366 | 0.6566 |
| Dasatinib | A04 | **0.6661** | 0.4796 | 0.5527 | 0.647 |
| Dasatinib | A05 | **0.7910** | 0.5014 | 0.5804 | 0.6904 |
| Dasatinib | A06 | **0.7979** | 0.5167 | 0.6258 | 0.6707 |
| Dasatinib | A07 | **0.8105** | 0.5215 | 0.6016 | 0.6809 |
| Dasatinib | A08 | **0.8047** | 0.6928 | 0.5802 | 0.633 |
| Dasatinib | A09 | **0.7485** | 0.5203 | 0.5958 | 0.6861 |
| Dasatinib | A10 | **0.8062** | 0.5158 | 0.5742 | 0.6503 |
| Dasatinib | A11 | **0.7837** | 0.5066 | 0.6331 | 0.6813 |
| GDC-0941 | A01 | 0.5632 | **0.6434** | 0.5987 | 0.6279 |
| GDC-0941 | A02 | **0.8257** | 0.7291 | 0.7349 | 0.7507 |
| GDC-0941 | A03 | **0.8268** | 0.7321 | 0.6822 | 0.7853 |
| GDC-0941 | A04 | **0.8389** | 0.7115 | 0.7569 | 0.7421 |
| GDC-0941 | A05 | **0.8382** | 0.7946 | 0.7171 | 0.7735 |
| GDC-0941 | A06 | **0.8463** | 0.6125 | 0.6858 | 0.764 |
| GDC-0941 | A07 | **0.8382** | 0.6061 | 0.7776 | 0.7612 |

| | | | | | |
|---|---|---|---|---|---|
| GDC-0941 | A08 | **0.8249** | 0.5493 | 0.6058 | 0.7796 |
| GDC-0941 | A09 | **0.8606** | 0.7689 | 0.8043 | 0.7206 |
| GDC-0941 | A10 | **0.8412** | 0.7227 | 0.653 | 0.6465 |
| GDC-0941 | A11 | 0.7859 | 0.5703 | 0.7297 | **0.7891** |
| GDC-0941 | A12 | **0.7803** | 0.6326 | 0.69 | 0.6727 |
| Go69 | A01 | 0.6520 | 0.6571 | **0.718** | 0.5822 |
| Go69 | A02 | **0.7835** | 0.7693 | 0.6075 | 0.7322 |
| Go69 | A03 | 0.7305 | 0.7334 | **0.757** | 0.6414 |
| Go69 | A04 | 0.7640 | 0.7456 | **0.8013** | 0.7425 |
| Go69 | A05 | **0.7812** | 0.7555 | 0.7294 | 0.7727 |
| Go69 | A06 | **0.7816** | 0.7404 | 0.7437 | 0.6443 |
| Go69 | A07 | 0.7407 | **0.8513** | 0.7527 | 0.6811 |
| Go69 | A08 | 0.7293 | **0.7338** | 0.6984 | 0.6525 |
| Go69 | A09 | **0.8228** | 0.6955 | 0.6985 | 0.7317 |
| Go69 | A10 | 0.7560 | 0.7512 | **0.7689** | 0.7071 |
| Go69 | A11 | **0.7565** | 0.7373 | 0.7213 | 0.7315 |
| Go69 | A12 | 0.7426 | 0.7086 | **0.7846** | 0.6442 |
| H89 | A01 | 0.6734 | **0.6952** | 0.6003 | 0.6105 |
| H89 | A02 | **0.7288** | 0.5391 | 0.5918 | 0.678 |
| H89 | A03 | **0.8051** | 0.5414 | 0.6856 | 0.6759 |
| H89 | A04 | **0.8144** | 0.7314 | 0.662 | 0.7287 |
| H89 | A05 | **0.7821** | 0.5468 | 0.6485 | 0.6672 |
| H89 | A06 | 0.7647 | 0.5636 | **0.8281** | 0.7165 |
| H89 | A07 | **0.7762** | 0.6983 | 0.7284 | 0.6442 |
| H89 | A09 | **0.8131** | 0.5442 | 0.5906 | 0.6707 |
| H89 | A10 | **0.7517** | 0.5549 | 0.6028 | 0.682 |
| H89 | A11 | **0.7417** | 0.7414 | 0.6863 | 0.7257 |
| H89 | A12 | **0.7939** | 0.6934 | 0.5831 | 0.6401 |
| IKKi | A02 | **0.7945** | 0.6619 | 0.7371 | 0.6475 |
| IKKi | A03 | 0.6873 | 0.6568 | 0.5661 | **0.6895** |
| IKKi | A04 | **0.7942** | 0.6754 | 0.6386 | 0.7052 |
| IKKi | A05 | **0.6977** | 0.6569 | 0.6157 | 0.6899 |
| IKKi | A06 | **0.7442** | 0.6931 | 0.7024 | 0.7077 |
| IKKi | A07 | **0.7352** | 0.5303 | 0.669 | 0.7001 |
| IKKi | A08 | **0.7470** | 0.7006 | 0.5358 | 0.6869 |
| IKKi | A09 | **0.8097** | 0.5175 | 0.6299 | 0.6969 |
| IKKi | A10 | **0.7647** | 0.6308 | 0.657 | 0.7334 |
| IKKi | A11 | **0.7878** | 0.6365 | 0.6757 | 0.6613 |
| IKKi | A12 | **0.6673** | 0.5629 | 0.497 | 0.6043 |
| Imatinib | A02 | **0.7935** | 0.7571 | 0.6721 | 0.7677 |
| Imatinib | A03 | **0.7763** | 0.7429 | 0.7041 | 0.7499 |
| Imatinib | A04 | **0.8058** | 0.7564 | 0.6921 | 0.7229 |
| Imatinib | A05 | **0.7714** | 0.7559 | 0.6689 | 0.7609 |
| Imatinib | A06 | **0.7756** | 0.746 | 0.6956 | 0.7296 |
| Imatinib | A07 | 0.7468 | **0.7515** | 0.6974 | 0.7137 |
| Imatinib | A08 | **0.7631** | 0.7534 | 0.5189 | 0.7096 |
| Imatinib | A09 | **0.8082** | 0.5605 | 0.5819 | 0.7447 |
| Imatinib | A10 | **0.7964** | 0.5645 | 0.5637 | 0.78 |
| Imatinib | A11 | 0.7289 | **0.7664** | 0.7576 | 0.7395 |
| Imatinib | A12 | 0.7012 | **0.8451** | 0.6369 | 0.7259 |
| Jak1i | A02 | **0.8210** | 0.5167 | 0.5771 | 0.616 |
| Jak1i | A03 | **0.7343** | 0.7139 | 0.6526 | 0.7133 |
| Jak1i | A04 | **0.7321** | 0.7066 | 0.6346 | 0.7189 |
| Jak1i | A05 | **0.7413** | 0.5163 | 0.6551 | 0.7089 |
| Jak1i | A06 | **0.7244** | 0.5525 | 0.6804 | 0.6905 |
| Jak1i | A07 | **0.7779** | 0.5499 | 0.5605 | 0.7099 |
| Jak1i | A08 | **0.7281** | 0.6995 | 0.6021 | 0.6605 |
| Jak1i | A09 | **0.8043** | 0.5064 | 0.6054 | 0.6717 |
| Jak1i | A10 | **0.7801** | 0.5295 | 0.5538 | 0.7015 |

| | | | | | |
|---|---|---|---|---|---|
| Jak1i | A11 | 0.7128 | 0.7307 | **0.7386** | 0.6812 |
| Jak1i | A12 | **0.7204** | 0.6229 | 0.6321 | 0.6905 |
| Jak2i | A01 | **0.6944** | 0.6379 | 0.6014 | 0.6207 |
| Jak2i | A02 | **0.7961** | 0.664 | 0.6656 | 0.7083 |
| Jak2i | A03 | **0.7629** | 0.6742 | 0.7138 | 0.7024 |
| Jak2i | A04 | **0.7890** | 0.6716 | 0.6227 | 0.7072 |
| Jak2i | A05 | **0.6666** | 0.4689 | 0.5314 | 0.6459 |
| Jak2i | A06 | **0.8110** | 0.6474 | 0.6651 | 0.6833 |
| Jak2i | A07 | **0.7595** | 0.6818 | 0.7593 | 0.6982 |
| Jak2i | A08 | **0.8050** | 0.6601 | 0.6152 | 0.686 |
| Jak2i | A09 | **0.8028** | 0.5253 | 0.6414 | 0.6501 |
| Jak2i | A10 | **0.8030** | 0.6762 | 0.6067 | 0.6364 |
| Jak2i | A11 | **0.8228** | 0.5398 | 0.694 | 0.7473 |
| Jak2i | A12 | **0.6831** | 0.6214 | 0.5825 | 0.5687 |
| Jak3i | A02 | **0.7986** | 0.7108 | 0.5666 | 0.6912 |
| Jak3i | A03 | **0.7170** | 0.7116 | 0.6991 | 0.7001 |
| Jak3i | A04 | **0.7983** | 0.5243 | 0.6654 | 0.691 |
| Jak3i | A05 | **0.7087** | 0.6498 | 0.6884 | 0.7073 |
| Jak3i | A06 | **0.7272** | 0.7244 | 0.654 | 0.7059 |
| Jak3i | A07 | **0.7768** | 0.5167 | 0.696 | 0.735 |
| Jak3i | A08 | 0.7196 | 0.6797 | 0.5946 | **0.7287** |
| Jak3i | A09 | **0.7988** | 0.6918 | 0.6013 | 0.6826 |
| Jak3i | A10 | **0.8026** | 0.7103 | 0.7104 | 0.7219 |
| Jak3i | A11 | **0.7281** | 0.5107 | 0.6854 | 0.6614 |
| Jak3i | A12 | **0.7511** | 0.6135 | 0.4861 | 0.61 |
| Lcki | A01 | 0.7359 | **0.7582** | 0.6106 | 0.7201 |
| Lcki | A02 | 0.7605 | 0.7453 | 0.6391 | **0.7696** |
| Lcki | A03 | **0.8032** | 0.5608 | 0.6814 | 0.721 |
| Lcki | A04 | 0.7608 | 0.5764 | 0.6788 | **0.7904** |
| Lcki | A05 | **0.8210** | 0.5435 | 0.7204 | 0.7442 |
| Lcki | A06 | 0.7564 | **0.7662** | 0.728 | 0.7556 |
| Lcki | A07 | **0.8304** | 0.579 | 0.6992 | 0.696 |
| Lcki | A08 | **0.7854** | 0.7457 | 0.5904 | 0.6972 |
| Lcki | A09 | **0.8452** | 0.5859 | 0.6018 | 0.7569 |
| Lcki | A10 | 0.7387 | **0.744** | 0.6598 | 0.6627 |
| Lcki | A11 | **0.7835** | 0.7639 | 0.6836 | 0.7558 |
| Lcki | A12 | 0.7467 | **0.8271** | 0.6888 | 0.6878 |
| PP2 | A02 | 0.7687 | 0.759 | **0.7717** | 0.7605 |
| PP2 | A03 | **0.8395** | 0.7644 | 0.7304 | 0.7953 |
| PP2 | A04 | **0.8442** | 0.7703 | 0.7116 | 0.7162 |
| PP2 | A05 | **0.8248** | 0.5777 | 0.7205 | 0.7547 |
| PP2 | A06 | **0.7866** | 0.7612 | 0.7461 | 0.7431 |
| PP2 | A07 | **0.8595** | 0.7616 | 0.724 | 0.7213 |
| PP2 | A08 | **0.8505** | 0.7489 | 0.7109 | 0.7195 |
| PP2 | A09 | **0.7902** | 0.5755 | 0.6511 | 0.7738 |
| PP2 | A10 | **0.8089** | 0.743 | 0.6635 | 0.7389 |
| PP2 | A11 | **0.7977** | 0.5852 | 0.6564 | 0.7846 |
| PP2 | A12 | **0.7667** | 0.6012 | 0.6524 | 0.6636 |
| Rapamycin | A01 | **0.7028** | 0.675 | 0.5882 | 0.5677 |
| Rapamycin | A02 | **0.7215** | 0.6831 | 0.6124 | 0.6697 |
| Rapamycin | A03 | **0.7322** | 0.6707 | 0.6296 | 0.6861 |
| Rapamycin | A04 | 0.6787 | 0.6696 | 0.6887 | **0.7267** |
| Rapamycin | A05 | **0.7231** | 0.653 | 0.7134 | 0.6466 |
| Rapamycin | A06 | **0.7310** | 0.6473 | 0.7009 | 0.6386 |
| Rapamycin | A07 | **0.7595** | 0.6642 | 0.748 | 0.5882 |
| Rapamycin | A08 | 0.7773 | **0.836** | 0.6371 | 0.571 |
| Rapamycin | A09 | **0.7732** | 0.6573 | 0.6826 | 0.6615 |
| Rapamycin | A10 | **0.7586** | 0.6702 | 0.7136 | 0.6344 |

| | | | | | |
|---|---|---|---|---|---|
| Rapamycin | A12 | **0.6955** | 0.6361 | 0.6561 | 0.5472 |
| SB202 | A01 | 0.6884 | 0.6713 | **0.941** | 0.7101 |
| SB202 | A03 | **0.7869** | 0.7549 | 0.6686 | 0.7633 |
| SB202 | A05 | **0.7856** | 0.5564 | 0.7387 | 0.6999 |
| SB202 | A06 | 0.7707 | 0.755 | **0.7913** | 0.7869 |
| SB202 | A10 | 0.7559 | 0.7554 | - | **0.7749** |
| SP6 | A01 | **0.7033** | 0.6882 | 0.4191 | 0.532 |
| SP6 | A02 | **0.7536** | 0.5035 | 0.5104 | 0.657 |
| SP6 | A03 | **0.7387** | 0.6973 | 0.534 | 0.5858 |
| SP6 | A04 | **0.6910** | 0.503 | 0.5065 | 0.5975 |
| SP6 | A05 | **0.7210** | 0.5068 | 0.5643 | 0.6869 |
| SP6 | A06 | 0.7052 | **0.719** | 0.5063 | 0.6384 |
| SP6 | A07 | **0.7281** | 0.7074 | 0.5382 | 0.6501 |
| SP6 | A08 | **0.7301** | 0.6832 | 0.4665 | 0.6133 |
| SP6 | A09 | **0.7743** | 0.5001 | 0.4618 | 0.6208 |
| SP6 | A10 | **0.7198** | 0.5111 | 0.524 | 0.6773 |
| SP6 | A11 | **0.7494** | 0.493 | 0.5407 | 0.5935 |
| SP6 | A12 | **0.7311** | 0.6131 | 0.4488 | 0.6198 |
| Sorafenib | A01 | 0.7185 | **0.7217** | 0.5884 | 0.6574 |
| Sorafenib | A02 | **0.8250** | 0.7659 | 0.6658 | 0.7664 |
| Sorafenib | A03 | 0.7689 | **0.7732** | 0.7078 | 0.6869 |
| Sorafenib | A04 | **0.8360** | 0.7094 | 0.7114 | 0.7218 |
| Sorafenib | A05 | **0.8304** | 0.5571 | 0.7672 | 0.7153 |
| Sorafenib | A06 | **0.8021** | 0.5783 | 0.6991 | 0.7506 |
| Sorafenib | A07 | **0.8461** | 0.7051 | 0.7267 | 0.6701 |
| Sorafenib | A09 | **0.8226** | 0.7275 | 0.7522 | 0.7587 |
| Sorafenib | A10 | **0.8103** | 0.7561 | 0.7457 | 0.7214 |
| Sorafenib | A11 | **0.8465** | 0.5777 | 0.7192 | 0.7503 |
| Sorafenib | A12 | **0.7715** | 0.6533 | 0.6084 | 0.6129 |
| Staurosporine | A01 | 0.7985 | **0.8464** | 0.6057 | 0.5945 |
| Staurosporine | A02 | **0.8347** | 0.8312 | 0.5999 | 0.6626 |
| Staurosporine | A03 | **0.8079** | 0.7072 | 0.6704 | 0.6787 |
| Staurosporine | A04 | 0.8418 | **0.8666** | 0.6452 | 0.6776 |
| Staurosporine | A05 | **0.8657** | 0.7305 | 0.7071 | 0.7515 |
| Staurosporine | A06 | **0.8694** | 0.516 | 0.6453 | 0.6619 |
| Staurosporine | A07 | **0.8277** | 0.7052 | 0.6349 | 0.6657 |
| Staurosporine | A08 | 0.8310 | **0.8316** | 0.6213 | 0.678 |
| Staurosporine | A09 | **0.8319** | 0.5117 | 0.6747 | 0.6726 |
| Staurosporine | A10 | **0.8417** | 0.5108 | 0.6211 | 0.7126 |
| Staurosporine | A11 | 0.8246 | **0.8711** | 0.6547 | 0.7445 |
| Streptonigrin | A01 | **0.7128** | 0.5689 | 0.6571 | 0.6599 |
| Streptonigrin | A02 | **0.7836** | 0.5095 | 0.549 | 0.6155 |
| Streptonigrin | A03 | **0.7776** | 0.547 | 0.6497 | 0.6527 |
| Streptonigrin | A04 | **0.8466** | 0.7521 | 0.5762 | 0.7061 |
| Streptonigrin | A05 | **0.8130** | 0.5406 | 0.6459 | 0.6928 |
| Streptonigrin | A06 | **0.8031** | 0.7409 | 0.6446 | 0.6343 |
| Streptonigrin | A07 | **0.7987** | 0.5353 | 0.5882 | 0.6657 |
| Streptonigrin | A08 | **0.7470** | 0.7458 | 0.5864 | 0.6443 |
| Streptonigrin | A09 | **0.7586** | 0.7034 | 0.5928 | 0.6196 |
| Streptonigrin | A10 | **0.7159** | 0.6974 | 0.5174 | 0.6809 |
| Streptonigrin | A11 | **0.8178** | 0.5649 | 0.593 | 0.6814 |
| Streptonigrin | A12 | **0.7410** | 0.6034 | 0.5896 | 0.6286 |
| Sunitinib | A01 | **0.7152** | 0.6622 | 0.5653 | 0.6522 |
| Sunitinib | A02 | **0.8056** | 0.498 | 0.6138 | 0.6521 |
| Sunitinib | A03 | **0.8095** | 0.6873 | 0.6889 | 0.6913 |
| Sunitinib | A04 | **0.8142** | 0.6925 | 0.6467 | 0.7121 |
| Sunitinib | A05 | **0.8157** | 0.6959 | 0.673 | 0.7073 |
| Sunitinib | A06 | **0.7968** | 0.5061 | 0.6654 | 0.7025 |
| Sunitinib | A07 | **0.8110** | 0.7 | 0.6333 | 0.6572 |

| | | | | | |
|---|---|---|---|---|---|
| Sunitinib | A08 | **0.8186** | 0.6894 | 0.5999 | 0.674 |
| Sunitinib | A09 | **0.8029** | 0.4886 | 0.6699 | 0.6621 |
| Sunitinib | A10 | 0.8126 | **0.848** | 0.6087 | 0.6713 |
| Sunitinib | A11 | **0.8241** | 0.824 | 0.6408 | 0.6811 |
| Sunitinib | A12 | 0.7747 | **0.7898** | 0.5942 | 0.5867 |
| Syki | A02 | **0.7682** | 0.7073 | 0.6636 | 0.685 |
| Syki | A03 | **0.7224** | 0.7042 | 0.6424 | 0.7116 |
| Syki | A04 | 0.7461 | 0.7069 | **0.7908** | 0.7256 |
| Syki | A05 | **0.7468** | 0.7182 | 0.6263 | 0.6804 |
| Syki | A06 | 0.7381 | 0.7134 | **0.7718** | 0.7154 |
| Syki | A07 | **0.7891** | 0.7 | 0.7434 | 0.6479 |
| Syki | A08 | **0.7509** | 0.7154 | 0.6903 | 0.6542 |
| Syki | A09 | **0.7712** | 0.73 | 0.7357 | 0.6918 |
| Syki | A10 | **0.7695** | 0.7531 | 0.7197 | 0.7242 |
| Syki | A11 | 0.7360 | 0.7311 | 0.7577 | **0.78** |
| Syki | A12 | 0.6717 | 0.6793 | **0.7426** | 0.7123 |
| U0126 | A01 | **0.6844** | 0.6178 | - | 0.6486 |
| U0126 | A02 | **0.8440** | 0.5545 | 0.5362 | 0.7043 |
| U0126 | A03 | **0.8340** | 0.5346 | 0.616 | 0.6881 |
| U0126 | A04 | **0.8263** | 0.7079 | 0.6166 | 0.7059 |
| U0126 | A05 | **0.8535** | 0.5468 | 0.7091 | 0.7031 |
| U0126 | A06 | **0.8199** | 0.5285 | 0.6018 | 0.6874 |
| U0126 | A07 | **0.8079** | 0.5304 | 0.5671 | 0.7249 |
| U0126 | A08 | **0.8278** | 0.6864 | 0.5359 | 0.6577 |
| U0126 | A09 | **0.8331** | 0.5394 | 0.5678 | 0.6967 |
| U0126 | A10 | **0.8436** | 0.5593 | 0.6092 | 0.6867 |
| U0126 | A11 | **0.7654** | 0.5072 | 0.6374 | 0.6767 |
| U0126 | A12 | **0.7227** | 0.6496 | 0.6253 | 0.6281 |
| VX680 | A01 | **0.6930** | 0.4818 | 0.6028 | 0.6452 |
| VX680 | A02 | **0.7340** | 0.711 | 0.5587 | 0.633 |
| VX680 | A03 | **0.7525** | 0.6976 | 0.5663 | 0.7292 |
| VX680 | A04 | **0.8127** | 0.6435 | 0.6722 | 0.5954 |
| VX680 | A05 | 0.6937 | 0.6742 | **0.7374** | 0.6454 |
| VX680 | A06 | **0.7168** | 0.7101 | 0.5769 | 0.6202 |
| VX680 | A07 | **0.7663** | 0.4944 | 0.5382 | 0.718 |
| VX680 | A08 | **0.7315** | 0.7082 | 0.4753 | 0.6482 |
| VX680 | A09 | **0.7703** | 0.7054 | 0.5859 | 0.6722 |
| VX680 | A10 | **0.7143** | 0.7137 | 0.6648 | 0.6167 |
| VX680 | A11 | 0.7050 | 0.6773 | **0.7269** | 0.6947 |
| VX680 | A12 | 0.7852 | **0.7922** | 0.5583 | 0.6808 |

