# OpenReview forum: "Mixture-of-Experts Variational Autoencoder for clustering and generating from similarity-based representations"
_ICLR.cc/2020/Conference — Reject_

### Official Review · AnonReviewer1 · 2019-10-23
**Official Blind Review #1**

**Rating:** 3

**Review:**

The proposed method of mixture-of-experts variational autoencoders
is valuable and insightful.
On the other hand the work could be improved and clarified at some points:

- in the abstract it is claimed that the method works for high-dimensional data.However, it should be better explained why this is the case. The method is largely based on density estimation with a mixture of Gaussians which is known to have limitations in higher dimensions (see e.g. classical textbooks like Bishop 1995)

- the similarity matrix and the similarity values should be carefully defined. Is there also an underlying similarity function assumed?

- a main shortcoming is that there is no discussion or experimental comparison with methods like spectral clustering and kernel spectral clustering. Given that the paper and the proposed method relates to similarity-based representations it would be important to know how it compares to such methods. Though e.g. in Table 1 the authors compare with about 10 other methods it would be more relevant that among some of these would have been spectral clustering and kernel spectral clustering, because of the similarity-based representations.

- in section 4.1 the MNIST data are taken with k=10. Though it is nicely explained and illustrated on this data set, it is possibly somewhat misleading as an example. The reason is that this is a classification problem with 10 classes, therefore the choice k=10 is obvious. It would be more important to consider benchmark problems for clustering, instead of classification, for which the choice of k is also an important model selection issue and for which k is unknown (how should k be selected then?).

- is each cluster always be assumed to be a Gaussian (which seems to be a strong assumption in general, and possibly not always realistic)? Could other components be used in the mixture?

**Experience Assessment:**

I have published one or two papers in this area.

**Review Assessment: Checking Correctness Of Derivations And Theory:**

I assessed the sensibility of the derivations and theory.

**Review Assessment: Checking Correctness Of Experiments:**

I assessed the sensibility of the experiments.

**Review Assessment: Thoroughness In Paper Reading:**

I read the paper at least twice and used my best judgement in assessing the paper.

---

> ### Author Response · Authors · 2019-11-09
> **Response to Review #1**
>
> Thank you very much for reviewing our paper. We addressed your concerns and suggestions as follows.  Reviewer comments are pasted and marked with ***, author responses follow below:
>
> ***- in the abstract it is claimed that the method works for high-dimensional data.However, it should be better explained why this is the case. The method is largely based on density estimation with a mixture of Gaussians which is known to have limitations in higher dimensions (see e.g. classical textbooks like Bishop 1995)***
>
> The reviewer is right about the issues of GMMs with high dimensional input. The input data for our model can readily be high-dimensional since the Variational Autoencoders have been demonstrated to be able to handle very well, i.e. to generate informative representations in a lower-dimensional latent space [Aljalbout et al. “Clustering with deep learning: Taxonomy and new methods”]. While being low-dimensional these representations capture the main differences in variance to be able to reconstruct the data. We fit our GMMs in this lower-dimensional latent space. We agree that one has to find a trade-off when choosing the dimensionality of the latent representation: it should be large enough to find separation and capture the variability of the input data, but also small enough such that the Gaussian mixtures can fit the clusters. We updated the manuscript to alert potential users about this issue and make this tradeoff clearer. We specifically address it in section 2 where we introduce the model and specify how we train the GMM on the latent representation. Specifically, for the MNIST application (d=68) and the real biological data (d=9) and we found good clustering performance where GMMs have been demonstrated to operate well [Jiang et al. “Variational deep embedding: An unsupervised and generative approach to clustering”].
>
> ————--
>
> ***- the similarity matrix and the similarity values should be carefully defined. Is there also an underlying similarity function assumed?***
>
> The similarity matrix indeed has to be defined carefully, since otherwise, the clustering network attempts to cluster data objects which might not be similar and therefore also the latent representation might not separate the clusters. The similarity matrices for each training batch in our experiments are defined either via k-nearest neighbors or via distance thresholds where the similarity function is the Euclidean distance. Both are applied to the transformed data using UMAP. To define the similarity matrix like that was the most straightforward way we could think of but could be easily replaced by a different approach in the MoE-Sim-VAE framework. The details are stated in the sections of the respective experiments. Further, we added an ablation study on the similarity matrix to our paper to section 4.2 and additionally with a figure (A4) in the appendix. Herewith, we show the importance and also the positive influence of the similarity matrix on the separation of the clusters in the latent representation. It shows a lower separation of the different classes in the latent representation when ignoring the similarity matrix when training the model.
>
> ————--
>
> ***- a main shortcoming is that there is no discussion or experimental comparison with methods like spectral clustering and kernel spectral clustering. Given that the paper and the proposed method relates to similarity-based representations it would be important to know how it compares to such methods. Though e.g. in Table 1 the authors compare with about 10 other methods it would be more relevant that among some of these would have been spectral clustering and kernel spectral clustering, because of the similarity-based representations.***
>
> Thanks for the pointer towards spectral clustering methods. We added results from a recent ICLR publication which performs spectral clustering in various forms on MNIST and compare accuracies and NMIs to the results of our model in Table 1 and show better performance with the MoE-Sim-VAE.

---

> > ### Author Response · Authors · 2019-11-09
> > **Response to Review #1**
> >
> > ————--
> >
> > ***- in section 4.1 the MNIST data are taken with k=10. Though it is nicely explained and illustrated on this data set, it is possibly somewhat misleading as an example. The reason is that this is a classification problem with 10 classes, therefore the choice k=10 is obvious. It would be more important to consider benchmark problems for clustering, instead of classification, for which the choice of k is also an important model selection issue and for which k is unknown (how should k be selected then?).***
> >
> > Choosing the parameter k=10 for the experiments on MNIST was mainly for purposes of fair comparison to the competitor methods, which used k=10 in their respective publications which are also explicitly clustering approaches. The main point of this experiment was to compare our method against the baselines on a popular benchmark problem, but we agree that the sensitivity of the method’s performance to the number of clusters is also a relevant question. The current manuscript already contains a few experiments for suitable applications on varying the parameter k. Specifically, we refer to section 4.1 where for synthetic data, generated from GMMs, we show the ability of our model to learn the correct number of clusters. In section 4.3 we report experiments on real biological data. For the biological data, we chose for all experiments k=25, an overestimation of the true number of cell populations in the data sets, and could still achieve good performance when comparing f-measures and therefore clustering results.
> >
> > ————--
> >
> > ***- is each cluster always be assumed to be a Gaussian (which seems to be a strong assumption in general, and possibly not always realistic)? Could other components be used in the mixture?***
> >
> > We fully agree that this is a limitation in the presented implementation of the method. A Gaussian mixture will not always be the best option for many datasets, which is why we also discuss in our conclusion that replacing the Gaussian mixture with different mixture distributions might be an interesting avenue for future work. Such extensions to the MoE-Sim-VAE are expected to be straightforward since our model does not make strict assumptions about the parametric form of the mixture distributions.
> >
> > We hope that this addresses your questions and concerns. If you have any other suggestions on how we could improve our paper, please do let us know.

---

### Official Review · AnonReviewer3 · 2019-10-25
**Official Blind Review #3**

**Rating:** 6

**Review:**

The authors present an extension of variational autoencoders (VAEs), where Gaussian distribution of the latent variable is replaced by a mixture of Gaussians. The approach can be used for clustering and generation. The authors carry out experiments to evaluate the performance of the method in these tasks and compare it to competing methods.

The paper is well written and easy to read and understand. Specialized related work is discussed. I find the extension of VAEs to GMMs interesting for the ICLR community, although it is somewhat straight forward in terms of its technical difficulty. However, the technical novelty together with the fine empirical evaluation are just good enough for ICLR, in my opinion.


**Experience Assessment:**

I have read many papers in this area.

**Review Assessment: Checking Correctness Of Derivations And Theory:**

I did not assess the derivations or theory.

**Review Assessment: Checking Correctness Of Experiments:**

I assessed the sensibility of the experiments.

**Review Assessment: Thoroughness In Paper Reading:**

I made a quick assessment of this paper.

---

> ### Author Response · Authors · 2019-11-09
> **Response to Review #3**
>
> Thank you very much for reviewing our paper and your overall positive feedback.

---

### Official Review · AnonReviewer2 · 2019-10-26
**Official Blind Review #2**

**Rating:** 6

**Review:**


Summary:

The paper proposes to expand the VAE architecture with a
mixture-of-experts latent representation, with a
mixture-component-specific decoder that can specialize in a specific
cluster.  Importantly, the method can take advantage of a similarity
matrix to help with the clustering.

Overall, I recommend a weak accept.  The method seems reasonable, and
the paper is well-written, but the results are only marginally better
than other methods, and there are several weaknesses with the proposed
architecture and experimental setup.

Positives:

* The idea of a more expressive variational distribution seems good,
  although it is not novel.

* The ability to have multiple decoder networks seems reasonable.

* The ability to incorporate domain knowledge (in the form of a
  similarity matrix S) is a plus.

* The experiments are thorough, although the method is generally only
  slightly better than competing methods.

Negatives:

* It's not clear if the similarity matrix S is already solving the
  clustering problem - in which case, why do we need the rest of the
  model?  For example, in your experiments you often used UMAP to
  cluster data.  How does using UMAP by itself work?  (Along these
  lines, it was not clear if your GMM experiments clustered data in
  the original space, or in the UMAP'd space - please clarify this).
  A good ablation would be to somehow remove the S matrix, to see if
  the model can accurately cluster samples.

* There is little variance in the generated samples.

* There is not a one-to-one mapping of clusters to labels, so it is
  hard to use this method to generate a specific type of data (for
  example, it is hard to generate a specific digit).  This is a big
  difference from, say, a conditional sampler as learned by a GAN.
  This also arises in Fig. 3, where it is clear that latent cluster
  assignments do not match human-interpretable cluster assignments.  I
  suppose this is to be expected, but taken with the previous point
  (little variance in generated samples) I think it seriously weakens
  the paper's claim that this is an "accurate an efficient data
  generation method."

* The method does not do well when the number of clusters is large.
  Regular GMMs seem to outperform it.

* I felt that this paper made excessive use of the appendix.  The
  paper is not self-contained enough, effectively violating the length
  restrictions.  Please make an effort to move key results back in to
  the main body of the paper.


Experiments to run:

An ablation regarding the similarity matrix S.

Clarification of whether GMM experiments are run in data-space, or
UMAP'd space.

MIXAE features prominently in your related works, but is not compared
to in your experiments.  It sounds like a natural comparison.  Please
run this experiment, or explain why it is not a comparable method.




**Experience Assessment:**

I have read many papers in this area.

**Review Assessment: Checking Correctness Of Derivations And Theory:**

I carefully checked the derivations and theory.

**Review Assessment: Checking Correctness Of Experiments:**

I carefully checked the experiments.

**Review Assessment: Thoroughness In Paper Reading:**

I read the paper thoroughly.

---

> ### Author Response · Authors · 2019-11-09
> **Response to Review #2**
>
> Thank you very much for reviewing our paper and your overall positive feedback. Reviewer comments are pasted and marked with ***, author responses follow below:
>
> **** It's not clear if the similarity matrix S is already solving the
>   clustering problem - in which case, why do we need the rest of the
>   model?  For example, in your experiments you often used UMAP to
>   cluster data.  How does using UMAP by itself work?  (Along these
>   lines, it was not clear if your GMM experiments clustered data in
>   the original space, or in the UMAP'd space - please clarify this).
>   A good ablation would be to somehow remove the S matrix, to see if
>   the model can accurately cluster samples.
>
> Experiments to run:
>
> An ablation regarding the similarity matrix S.***
>
> The similarity matrix in combination with the clustering network is a key component of the model. We followed your advice to assess in more detail whether the similarity matrix already solves the clustering problem and performed an ablation study on the similarity matrix. We did so by rerunning the experiment on MNIST and setting the loss coefficient for L_Similarity to zero and thereby effectively removing the similarity matrix from our model. As a result, we observed that the clustering network is not able to perform the clustering anymore, mainly because the separation of the different classes in the latent representation is heavily impaired. We added the results to the revised version of our manuscript to section 4.2 and additionally with a figure (A4) in the appendix.
>
> ————--
>
> ***Clarification of whether GMM experiments are run in data-space, or
> UMAP'd space.***
>
> Similarly, as in our model, we run the GMM experiments also on the data space. The UMAP projection is only used as a transformation and similarity measure to include domain knowledge encoded in the similarity matrix and therefore also to show the advantage of our model.
>
> ————--
>
> **** There is not a one-to-one mapping of clusters to labels, so it is
>   hard to use this method to generate a specific type of data (for
>   example, it is hard to generate a specific digit).  This is a big
>   difference from, say, a conditional sampler as learned by a GAN.
>   This also arises in Fig. 3, where it is clear that latent cluster
>   assignments do not match human-interpretable cluster assignments.  I
>   suppose this is to be expected, but taken with the previous point
>   (little variance in generated samples) I think it seriously weakens
>   the paper's claim that this is an "accurate an efficient data
>   generation method."***
>
> We fully agree with your feedback on the one-to-one mapping between cluster-ID and labels. This is indeed a disadvantage in comparison to the conditional GANs you mentioned. Future work could incorporate a feature in the framework to be able to condition on the label for generation purposes. Nevertheless, we believe that currently with little effort one can interpret the latent representation and therefore also find a mapping between the experts and label IDs.
>
> ————--
>
> **** There is little variance in the generated samples. ***
>
> We discuss this issue in the conclusion. It is a known “problem” with VAE that the sample variances and also the sharpness of generated images is lower in comparison to for example images generated with GANs [Dumoulin et al. “Adversarially Learned Inference”;  Theis et al. “A note on the evaluation of generative models”]. We expect that adding adversarial training could be a remedy to generate even more realistic samples as discussed in the conclusion.
>
> ————--
>
> ***MIXAE features prominently in your related works, but is not compared
> to in your experiments.  It sounds like a natural comparison.  Please
> run this experiment, or explain why it is not a comparable method.***
>
> Due to an oversight on our end, we missed to include the results from MIXAE, which we discuss in our related work section. We added them to the manuscript now (Table 1).
>
> We hope that this addresses your questions and concerns. If you have any other suggestions on how we could improve our paper, please do let us know.

---

### Decision · Program_Chairs · 2019-12-19

**Decision:**

Reject

**Comment:**

The paper proposes a VAE with a mixture-of-experts decoder for clustering and generation of high-dimensional data. Overall, the reviewers found the paper well-written and structured , but in post rebuttal discussion questioned the overall importance and interest of the work to the community.  This is genuinely a borderline submission. However, the calibrated average score currently falls below the acceptance threshold, so I’m recommending rejection, but strongly encouraging the authors to continue the work, better motivating the importance of the work, and resubmitting.